J Physiol 601.18 (2023) pp 4013–4032

4013

# What determines the optimal pharmacological treatment of atrial fibrillation? Insights from *in silico* trials in 800 virtual atria

Albert Dasí[1] , Michael T.B. Pope[2,3], Rohan S. Wijesurendra[2,4] , Tim R. Betts[2], Rafael Sachetto[5] , Alfonso Bueno-Orovio[1] and Blanca Rodriguez[1]

[1]*Department of Computer Science, University of Oxford, Oxford, UK*
[2]*Department of Cardiology, Oxford University Hospitals NHS Foundation Trust, Oxford, UK*
[3]*Department for Human Development and Health, University of Southampton, Southampton, UK*
[4]*Oxford Centre for Clinical Magnetic Resonance Research, Division of Cardiovascular Medicine, Radcliffe Department of Medicine, University of Oxford, Oxford, UK*
[5]*Departamento de Ciência da Computação, Universidade Federal de São João del-Rei, São João del-Rei, Brazil*

Handling Editors: Harold Schultz & Eleonora Grandi

The peer review history is available in the Supporting Information section of this article (https://doi.org/10.1113/JP284730#support-information-section).

The Journal of Physiology

**Abstract** The best pharmacological treatment for each atrial fibrillation (AF) patient is unclear. We aim to exploit AF simulations in 800 virtual atria to identify key patient characteristics that guide the optimal selection of anti-arrhythmic drugs. The virtual cohort considered variability in electrophysiology and low voltage areas (LVA) and was developed and validated against experimental and clinical data from ionic currents to ECG. AF sustained in 494 (62%) atria, with large inward rectifier K$^+$ current ($I_{K1}$) and Na$^+$/K$^+$ pump ($I_{NaK}$) densities ($I_{K1}$ 0.11 ± 0.03 *vs.* 0.07 ± 0.03 S mF$^{-1}$; $I_{NaK}$

$0.68 \pm 0.15$ *vs*. $0.38 \pm 26$ S mF$^{-1}$; sustained *vs*. un-sustained AF). In severely remodelled left atrium, with LVA extensions of more than 40% in the posterior wall, higher $I_{K1}$ (median density $0.12 \pm 0.02$ S mF$^{-1}$) was required for AF maintenance, and rotors localized in healthy right atrium. For lower LVA extensions, rotors could also anchor to LVA, in atria presenting short refractoriness (median L-type Ca$^{2+}$ current, $I_{CaL}$, density $0.08 \pm 0.03$ S mF$^{-1}$). This atrial refractoriness, modulated by $I_{CaL}$ and fast Na$^+$ current ($I_{Na}$), determined pharmacological treatment success for both small and large LVA. Vernakalant was effective in atria presenting long refractoriness (median $I_{CaL}$ density $0.13 \pm 0.05$ S mF$^{-1}$). For short refractoriness, atria with high $I_{Na}$ (median density $8.92 \pm 2.59$ S mF$^{-1}$) responded more favourably to amiodarone than flecainide, and the opposite was found in atria with low $I_{Na}$ (median density $5.33 \pm 1.41$ S mF$^{-1}$). *In silico* drug trials in 800 human atria identify inward currents as critical for optimal stratification of AF patient to pharmacological treatment and, together with the left atrial LVA extension, for accurately phenotyping AF dynamics.

(Received 22 March 2023; accepted after revision 5 July 2023; first published online 20 July 2023)

**Corresponding authors** A. Dasí and B. Rodriguez: Department of Computer Science, University of Oxford, Wolfson Building, Parks Road OX1 3QD Oxford, UK. Email: albert.dasiimartinez@cs.ox.ac.uk and blanca.rodriguez@cs.ox.ac.uk

**Abstract figure legend** Decision algorithm for optimal stratification of virtual atrial fibrillation patients to anti-arrhythmic drug therapy.

## Key points

- Atrial fibrillation (AF) maintenance is facilitated by small L-type Ca$^{2+}$ current ($I_{CaL}$) and large inward rectifier K$^+$ current ($I_{K1}$) and Na$^+$/K$^+$ pump.
- In severely remodelled left atrium, with low voltage areas (LVA) covering more than 40% of the posterior wall, sustained AF requires higher $I_{K1}$ and rotors localize in healthy right atrium. For lower LVA extensions, rotors can also anchor to LVA, if the atria present short refractoriness (low $I_{CaL}$)
- Vernakalant is effective in atria presenting long refractoriness (high $I_{CaL}$). For short refractoriness, atria with fast Na$^+$ current ($I_{Na}$) up-regulation respond more favourably to amiodarone than flecainide, and the opposite is found in atria with low $I_{Na}$.
- The inward currents ($I_{CaL}$ and $I_{Na}$) are critical for optimal stratification of AF patient to pharmacological treatment and, together with the left atrial LVA extension, for accurately phenotyping AF dynamics.

## Introduction

Pharmacological therapy remains a core element of atrial fibrillation (AF) management (Heijman, Hohnloser & John Camm, 2021). Anti-arrhythmic drugs, however, have moderate efficacy, substantial recurrence rates and undesired proarrhythmic effects (Heijman et al., 2018).

The situation worsens with AF progression, with AF eventually becoming refractory to therapy (Heijman, Linz & Schotten, 2021). Improving pharmacological treatment, through tailored therapies that consider individual AF mechanisms and interpatient characteristics (Heijman et al., 2018; Heijman, Hohnloser & John Camm, 2021), is therefore paramount. In this sense, human modelling

**Albert Dasí** is a third-year PhD student in the Computational Cardiovascular Science Team at the University of Oxford. Albert joined Professor Blanca Rodriguez's laboratory as part of a Marie Skłodowska-Curie Innovative Training Network (PersonalizeAF), with the aim of developing tools for the systematic analysis of novel and existing treatments for atrial fibrillation. His ongoing project focuses on conducting *in silico* trials (i.e. assessment of clinical trials in populations of virtual patient models) for a better stratification of patients to optimal therapies. His future aspiration is to integrate computational modelling and simulation into clinical research to advance precision medicine in atrial fibrillation.

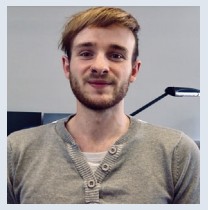

and simulation represent an effective means for studying treatment personalization (Heijman, Hohnloser & John Camm, 2021). Besides being considered a central novel paradigm for drug safety evaluation (Passini et al., 2017), *in silico* trials hold potential for unravelling key modulators of successful pharmacological therapy (Margara et al., 2022). They offer the possibility of testing several AF treatments in large cohorts of human atria, addressing the bias associated with human variability. Likewise, the inclusion of interpatient variability (i.e. atrial electrophysiology, tissue structure, etc.) allows the identification of optimal therapies for specific patient profiles.

The structurally-remodelled substrate has been widely associated with higher AF recurrence rates (Jadidi et al., 2018). This arrhythmogenic remodelling can be identified using voltage mapping (Jadidi et al., 2018) because the bipolar voltage is considered a marker of atrial (dys) function (Miyamoto et al., 2009). Low voltage areas (LVA, bipolar voltage lower than 0.5 mV in sinus rhythm) (van Schie et al., 2021) represent regions of atrial damage, often characterized by conduction abnormalities (Kishima et al., 2020), prolonged atrial activation times, slowed conduction velocity (van Schie et al., 2021) and electrogram fractionation (Miyamoto et al., 2009). LVA are primarily identified in late stages of AF (Masuda et al., 2020), but might also be present in paroxysmal AF patients (Masuda et al., 2018).

Although LVA are considered to have a central role on AF maintenance, contrasting results have been reported after ablating LVA in addition to pulmonary vein isolation (Junarta et al., 2022). Initial studies observed that LVA ablation yielded a greater arrhythmia freedom in persistent AF patients (Yang et al., 2017). However, the latest trials found no beneficial impact of LVA ablation in either paroxysmal (Masuda et al., 2020) or persistent (Yang et al., 2022) AF. Thus, the role of LVA on AF dynamics remains poorly understood, and little is known about how the patient's ionic current substrate combines with LVA to perpetuate AF. Indeed, whereas previous simulation studies argued that AF maintenance required fibrotic tissue (McDowell et al., 2015; Morgan et al., 2016; Zahid et al., 2016), sustained AF was commonly observed in human patients without structural heart disease (Kumar et al., 2012). Thus, the interaction of the ionic current properties with LVA, rather than the structural substrate alone, might be the core determinant of AF dynamics. Most importantly, because interpatient variability in ionic current densities is expected to influence the patient response to anti-arrhythmic drugs (Capucci et al., 2018), the former represents the cornerstone of pharmacological treatment personalization.

Thus, the present study aimed to perform *in silico* trials in a large cohort of human atria, with variability in the extent of LVA and membrane kinetics, to identify key determinants of AF treatment that could guide the decision-making of optimal pharmacological therapies. For this, AF was simulated in a population of 800 human atria, and analysed through ECG metrics and high-frequency maps. *In silico* trials were conducted with 12 pharmacological treatments used in the clinic, and simulations underwent consistent experimental and clinical validation. Our results provide evidence that the characterization of the exact pattern of atrial scarring and the ionic current substrate may be required for successfully phenotyping AF and personalizing pharmacological therapies.

## Methods

### Population of virtual atria models

A population of 800 virtual atria was developed combining a range of LVA distributions and atrial ionic current profiles. Figure 1 illustrates the overall process and an explanation is provided below.

**Patient-specific bipolar voltage maps.** Left atrial bipolar voltage maps were obtained from eight AF patients presenting high variability in the distribution, location and extent of LVA. High density mapping of the atria was performed using an Advisor HD grid (SE) (Abbott Vascular, Chicago, IL, USA) catheter when pacing at the coronary sinus at a cycle length of 800 ms. The study was conducted according to the principles of the *Declaration of Helsinki* and all patients provided their written informed consent (clinical trial identifier: NCT04229472).

**Registration of the patient-specific voltage maps to a human-based whole-atria model.** The left atrial surface of each patient was registered to the left atrium of the whole-atria model described in Seemann et al. (2006) through a rigid and non-rigid registration (White et al., 2019). The voltage data were subsequently interpolated using the nearest neighbor algorithm. Thus, eight human whole-atria models, presenting LVA variability were constructed based on the patient-specific maps.

**Modelling of low voltage areas.** Each whole-atria model was binarized into non-LVA and LVA (i.e. bipolar voltage lower than 0.5 mV). LVA were modelled as uniform regions (i.e. no difference between scar and border-zone) of 30% decreased conductivity, increased anisotropy (i.e. 8:1 longitudinal to transversal ratio) and ionic current dysregulation, resembling the effects of the transforming growth factor-$\beta$ (Roney et al., 2022). The latter consisted of a 50%, 40% and 50% decrease of the L-type $Ca^{2+}$ current density ($G_{CaL}$), fast $Na^{+}$ current density ($G_{Na}$) and inward

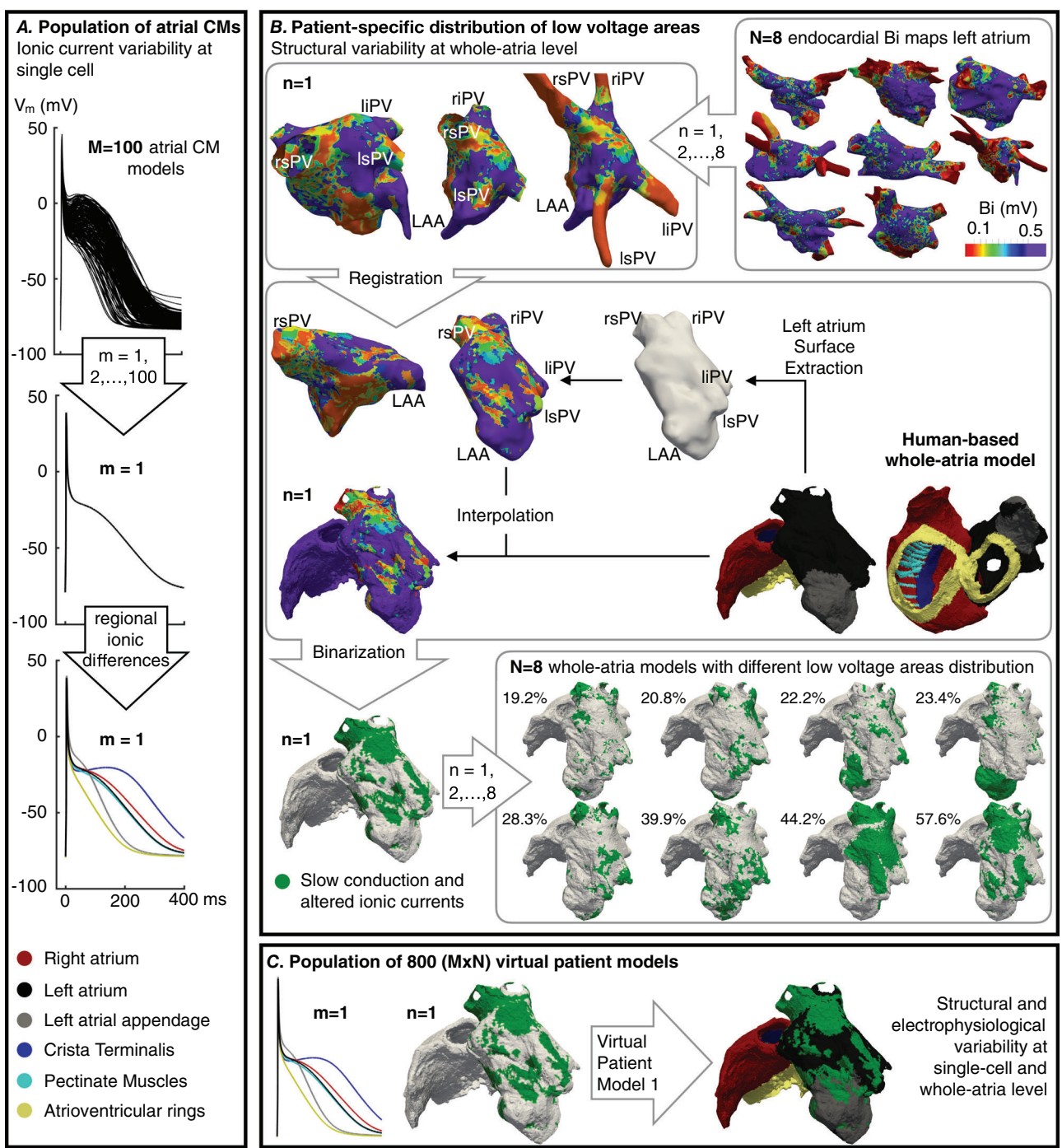

**Figure 1. Generation of the population of 800 virtual atria**

*A*, M = 100 atrial cardiomyocyte (CM) models reflecting variability in ionic current densities are developed and scaled in different atrial regions. *B*, *N* = 8 bipolar voltage (Bi) maps of the left atrium are registered to a human-based whole-atria model. The resulting whole-atria models are binarized into low volage areas (LVA, Bi < 0.5 mV) and non-LVA. Abbreviations. rs/i-PV, ls/i-PV, right superior/inferior, left superior/inferior pulmonary veins; LAA, left atrial appendage. *C*, a final population of 800 virtual patient models is generated combining each cardiomyocyte model, scaled in different atrial regions, with each whole-atria model with a specific LVA distribution. [Colour figure can be viewed at wileyonlinelibrary.com]

rectifier $K^+$ current density ($G_{K1}$), respectively. The LVA remodelling was applied on top of the individual electrophysiological properties of each whole-atria model (see 'Population of whole-atria models' below).

**Left atrium parcellation.** To identify preferential regions of LVA infiltration, the left atrium was subdivided into 13 segments: 1−4, posterior; 5−6, inferior; 7, septal; 8−11 anterior; 12, lateral wall (Benito et al., 2018); and 13, appendage. The right atrium, Bachmann Bundle and interatrial connections were modelled as non-LVA, and therefore not included in the parcellation, to isolate the role of structurally-remodelled left atrium on AF dynamics and pharmacological treatment.

**Population of atrial cardiomyocyte models.** To reflect variability in the patient electrophysiological substrate, each atrial model considered a specific ionic current profile. As previously (Dasí et al., 2022), a population of atrial cardiomyocyte models was generated with the CRN model (Courtemanche, Ramirez & Nattel, 1998), by varying current densities up to ±50% (Muszkiewicz et al., 2018). The ultrarapid, rapid and slow delayed-rectifier $K^+$ current ($G_{Kur}$, $G_{Kr}$ and $G_{Ks}$), transient outward $K^+$ current ($G_{to}$), $G_{K1}$, $G_{CaL}$, $G_{Na}$, $Na^+/K^+$ pump ($G_{NaK}$), $Ca^{2+}/Na^+$ exchanger ($G_{NCX}$), and the sarcoplasmic reticulum $Ca^{2+}$ release ($G_{rel}$), leak ($G_{leak}$) and uptake ($G_{up}$) currents were included.

Three-hundred cardiomyocyte models, generated with Latin hypercube sampling, were calibrated against experimental data obtained from patients in sinus rhythm and AF (Sánchez et al., 2014). A representative sample of the overall ionic current variability, consisting of 100 atrial cardiomyocyte models, was randomly selected.

**Population of whole-atria models.** The final population of 800 virtual atria was built combining the 100 atrial cardiomyocyte models with the eight LVA distributions. The single-cell properties of each atrial cardiomyocyte model were assigned to the left atrial tissue and modified in right atrium, crista terminalis, pectinate muscles, left atrial appendage and atrioventricular rings (a detailed explanation can be found in Dasí et al., 2022). Regional heterogeneities in conduction velocity and anisotropy ratio were similarly considered (Dasí et al., 2022; Sánchez et al., 2017), setting the longitudinal velocity in the bulk tissue to 80 cm s$^{-1}$, according to patient data.

Once the electrophysiological properties were defined in the different atrial regions, the LVA remodelling was implemented in structurally-impaired areas. Although this remodelling was consistently applied across models (i.e. always 50%, 60% and 50% reduction in $G_{CaL}$, $G_{Na}$ and $G_{K1}$, respectively), heterogeneous effects arose in both intra- and interatrial models. Regarding interatrial model heterogeneities, different atria presented different ionic current densities. Therefore, even the same 50-60-50% reduction in $G_{CaL}$-$G_{Na}$-$G_{K1}$ had differential effects on each model. Similarly, LVA infiltration could spread throughout the left atrial body, appendage, septum or atrioventricular rings. Because these atrial structures had heterogeneous electrophysiological properties (i.e. intra-atrial differences), applying the same LVA remodelling yielded an heterogenous ionic current dysregulation within atrial regions. Similar effects occurred for the conduction velocity and anisotropic ratio.

### AF inducibility

AF was induced in virtual atria by imposing spiral wave re-entries as the initial conditions of the simulation, and AF dynamics were analysed for 7 s of activity (Matene & Jacquemet, 2012; Roney et al., 2022). Six spiral waves re-entries were imposed in the atria (Matene & Jacquemet, 2012), with three in each atrial chamber. In the left atrium, one spiral wave was induced in the anterior wall, one in the posterior wall and another in the inferior wall (equally distributed in space). Two spiral waves were induced in the venous portion of the right atrium, one around the proximity of the superior cava vein and one close to the inferior cava vein, and a third one in the inferior wall. The direction of rotation was clockwise for two spiral waves in each chamber and counter clockwise in the third one, ensuring that adjacent re-entries rotated with opposing phase (Matene & Jacquemet, 2012). The sensitivity of AF induction by spiral-wave re-entries is addressed in the Discussion.

The three-dimensional monodomain equation of the transmembrane voltage was solved using MonoAlg3D (Sachetto Oliveira et al., 2018).

### ECG

Simulated eight-lead (leads I, II, V1–V6) ECGs were computed for sustained (>7 s) AF episodes as described in Gima & Rudy (2002). Five biomarkers, namely, dominant frequency, organization index, Shannon's spectral entropy, sample entropy and relative harmonic energy, were extracted from each lead as described in Alcaraz et al. (2011); Zeemering et al. (2018).

### Dominant frequency maps

Dominant frequency maps were calculated in sustained (>7 s) AF episodes as described in Sánchez et al. (2017). The maps were discretized into three frequency values: low (10th percentile), middle (10–90th percentile) and high frequency (90th percentile). High-frequency regions

**Table 1. Ionic current block (%) exerted by the anti-arrhythmic drugs modelled in the present study**

| Drug | | Dose | Ionic current block (%) | | | | | | | | | Reference |
|------|--|------|-------|------|------|------|------|-------|-------|-------|------|-----------|
| | | | $I_{Kur}$ | $I_{Kr}$ | $I_{to}$ | $I_{K1}$ | $I_{Ks}$ | $I_{CaL}$ | $I_{NaK}$ | $I_{NCX}$ | $I_{Na}$ | |
| Amiodarone | Acute | 1.5 µM | – | 40 | – | – | 30 | 30 | – | 30 | 20 | Bai, Lu & Zhang (2020); Loewe et al. |
| | | 3.0 µM | – | 65 | 30 | 20 | 50 | 65 | 10 | 50 | 40 | (2014); Sutanto et al. (2019); Varela |
| | Chronic | 3.0 µM | – | 65 | 30 | 20 | 50 | 0 | 10 | 50 | 40 | et al. (2016) |
| | | 3.0 µM | – | 65 | 30 | 40 | 50 | 0 | 25 | 50 | 40 | (Patel, Yan & Kowey, 2009) |
| | | 3.0 µM | – | 65 | 30 | 50 | 50 | 0 | 25 | 50 | 40 | |
| Flecainide | | 1.0 µM | – | 30 | – | – | – | – | – | – | 40 | (Bai et al., 2020; Jin et al., 2022; |
| | | 2.0 µM | 30 | 60 | 15 | – | – | 10 | – | – | 60 | Sanchez de la Nava et al., 2021; |
| | | 2.0 µM | 30 | 60 | 15 | – | – | 10 | – | – | 70 | Sutanto et al., 2019) |
| Vernakalant | | 10 µM | 40 | 30 | 40 | – | – | – | – | – | 10 | Loewe et al. (2015); Sutanto et al. |
| | | 30 µM | 70 | 60 | 60 | – | – | 20 | – | – | 30 | (2019); Varela et al. (2016) |
| Digoxin | | 5 nM | – | 40 | – | – | – | – | 30 | – | – | Bai, Lu & Zhang (2020) |
| | | 10 nM | – | 50 | – | – | – | – | 60 | – | – | |

obtained for a given LVA distribution with different ionic profiles were superimposed to generate frequency density maps. The latter maps identified preferential regions of high atrial activity (i.e. AF drivers) for each LVA distribution.

### *In silico* drug trials

Virtual atria with sustained (>7 s) AF were subjected to the administration of four drugs commonly used for AF treatment (Hindricks et al., 2021). Their administration was modelled 2 s from AF initiation and the episode was continued until completion of the original 7 s (i.e. 2 s in the absence of intervention and 5 s after virtual drug administration) (Matene & Jacquemet, 2012). Pharmacological AF prevention was considered successful if the atria were free of arrhythmic activity before the conclusion of the 7 s. Drug efficacy was defined as the percentage of virtual patients without sustained AF after drug administration.

For each drug, different doses were administered within the therapeutic plasma concentration range: vernakalant 10 and 30 µM (Roy et al., 2004), amiodarone 1.5 and 3.0 µM (Nishimura, Follmer & Singer, 1989), flecainide 1 and 2 µM (Viswanathan et al., 2001), and digoxin 5 and 10 nM (Chamberlain et al., 1970). Digoxin, clinically used for rate control, was also considered for rhythm control of AF because $I_{NaK}$ inhibition might hold potential for rotor termination (Bueno-Orovio et al., 2014; Dasí et al., 2022). Flecainide 2 µM considered the high levels of $I_{Na}$ blockade observed at fast activation rates ($\sim$ 60–70% blockade at 10 Hz) (Moreno et al., 2011). Acute and chronic amiodarone 3.0 µM were modelled with and without $I_{CaL}$ inhibition, respectively. Chronic amiodarone additionally comprised a progressive blockade of the Na$^+$/K$^+$ pump (Grey et al., 1998) and inward K$^+$ current

(Guillemare et al., 2000). Drug action was simulated as simple pore-block models because the data required were available for the compounds investigated. Table 1 collects the ionic current blockade of each treatment.

### Statistical analysis

Data normality was assessed by the Kolmogorov–Smirnov test. Non-parametric data are presented as the median and interquartile range (unless otherwise indicated) and analysed using the Wilcoxon rank sum test. $P < 0.05$ was considered statistically significant.

However, simulation studies go beyond the use of statistical tests and include a comprehensive evaluation of the physiological significance of the findings, regardless of specific $P$ values (Liberos et al., 2017). This represents the advantage of using populations of models calibrated and validated with experimental and clinical data for gaining mechanistic insights into cardiovascular variability, in that the effective size obtained from simulated data can be used as estimates for power calculations.

## Results

### AF sustains in virtual atria presenting short refractoriness and high cellular excitability, with the latter proportional to the extent of LVA in the left atrium

Sustained AF (>7 s) developed in 494 (62%) of the 800 virtual atria. Figure 2*A* illustrates a representative AF episode and the corresponding ECG. Simulated and clinical recordings shared similar morphology and complexity of fibrillatory-waves (f-waves), as well as comparable AF dominant frequency, supporting the credibility of the simulation results. Figure 2*B*

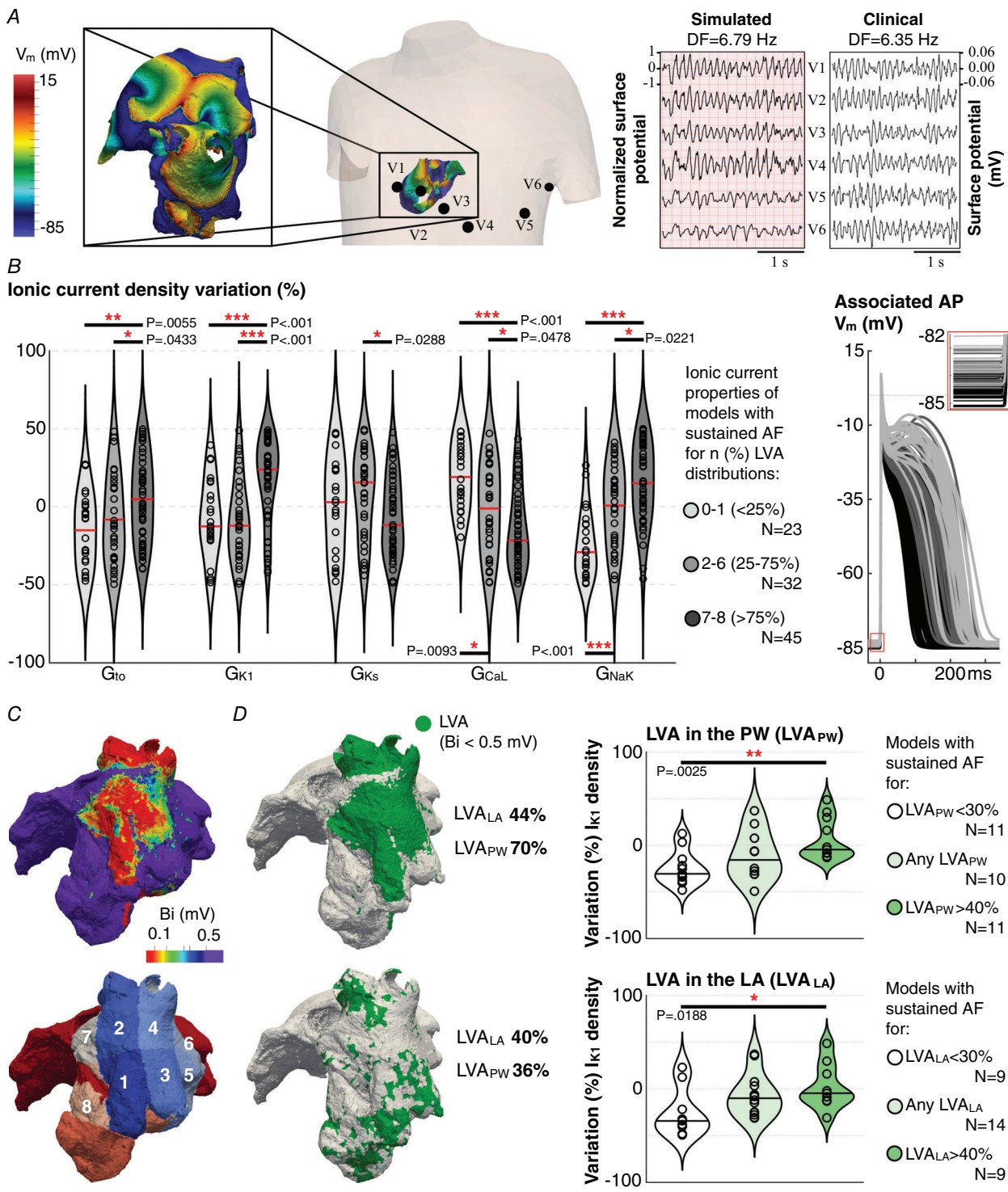

**Figure 2. Ionic current substrate favouring atrial fibrillation (AF) for distinct low voltage areas (LVA)**
*A*, snapshot of the atrial transmembrane voltage map during AF and location of the atria within the torso (left). Simulated and clinical ECG during AF and dominant frequency (DF) of lead V1 [right, adapted with permission from Lankveld et al. (2016)]. The clinical ECG was recorded with a sampling frequency of 250 Hz and postprocessing consisted of a zero-phase band-pass filter between 1 and 100 Hz, a 50 Hz notch filter and cancellation of ventricular signals: QRST-waves. Further details are available in Lankveld et al. (2016). *B*, variation of $G_{to}$, $G_{K1}$, $G_{Ks}$, $G_{CaL}$ and $G_{NaK}$ (with respect to baseline) yielding sustained AF (>7 s) for an increasing number of LVA distributions (left), and associated atrial action potential (AP, right). *C*, bipolar voltage (Bi) map and left atrium parcellation: Estimation of LVA in the left atrium posterior wall (regions 1–4). *D*, $I_{K1}$ density variation leading to sustained AF for an increasing LVA extension in the left atrium (LVA$_{LA}$, bottom) and in the left atrial posterior wall (LVA$_{PW}$, top). [Colour figure can be viewed at wileyonlinelibrary.com]

characterizes the atrial ionic current substrate favoring AF maintenance. The 100 ionic current profiles included in the present study have been grouped into three categories: profiles sustaining AF (>7 s) for less than two LVA distributions (less than 25% of the LVA distributions studied), between two and six (25–75% of the LVA distributions) and for more than six LVA distributions (more than 75% of the LVA distributions). Figure 2*C* and *D* shows the left atrium parcellation and the ionic current substrate needed for AF maintenance as the LVA extension increased.

Twelve ionic current profiles led to unsustained AF in all LVA distributions, whereas 11 perpetuated AF in only one out of the eight (less than 25% of the LVA distributions considered). Ionic profiles leading to low AF sustenance were characterized by significantly high $G_{CaL}$ and low $G_{NaK}$ compared to baseline (i.e. ionic current density variation equal to 0%). $G_{NaK}$ increase and $G_{CaL}$ decrease lead to AF developing in more LVA distributions, with the 45 most proarrhythmic profiles (i.e. sustaining AF for more than 75% of the LVA distributions) showing the highest $G_{NaK}$ and the lowest $G_{CaL}$. The 45 AF-prone ionic current substrates additionally presented $G_{to}$ and $G_{K1}$ up-regulation, which yielded a shorter refractory period ($185.3 \pm 56.7$ *vs.* $125.3 \pm 18.2$ ms; atria with sustained AF for less than 25% *vs.* more than 75% LVA distributions) and a more negative resting membrane potential ($-83.6 \pm 0.4$ *vs.* $-84.5 \pm 0.4$ mV) (Fig. 2*B*).

$I_{K1}$ up-regulation was especially needed as the LVA density increased in the left atrium. LVA were characterized by low cellular excitability (i.e. conduction impairment and reduced $I_{Na}$, $I_{CaL}$ and $I_{K1}$). When they occupied a large portion of the left atrium (i.e. extending throughout the entire posterior wall >40% LVA) (Fig. 2*D*), sustained AF (>7 s) could only result from ionic current profiles exhibiting the highest $I_{K1}$. This is illustrated in the 32 ionic current profiles yielding AF maintenance in two to six LVA distributions. From the 32 profiles, nine sustained AF (>7 s) primarily in smaller LVA extensions (20–30% LVA), nine in larger (40–60% LVA) and 14 in both. Compared to ionic current profiles sustaining AF in smaller LVA, those profiles engaging with severe structural remodelling showed a significantly higher $I_{K1}$. This difference was accentuated when considering the LVA extension in the left atrial posterior wall (Fig. 2*C*, regions 1–4).

simulated ECGs were computed in the 494 sustained AF (>7 s) episodes. Table 2 shows the influence of ionic current dysregulation on five ECG markers (i.e. dominant frequency, organization index, spectral entropy, sample entropy and relative harmonic energy).

Overall, dysregulation of $I_{CaL}$, $I_{to}$, $I_{Na}$, $I_{NaK}$ and $I_{K1}$ was associated with variations in ECG metrics (Table 2). For some LVA distributions, high $G_{CaL}$ and low $G_{to}$ increased the ECG spectral entropy. The spectral entropy relates to the uniformity of fibrillation frequencies and a high value indicates high AF complexity (Zeemering et al., 2018). Elevated $G_{CaL}$ was associated with APD alternans and wavefront fractionation (Dasí et al., 2022), consistent with an increase in AF complexity and the spectral entropy. Conversely, low $G_{CaL}$ and high $G_{to}$ were associated with an elevation of the sample entropy, a measure of predictability and repetitiveness (Zeemering et al., 2018). As such, $G_{CaL}$ down-regulation yielded the formation of localized rotors (see next section below), which increased AF repetitiveness. High $G_{NaK}$ was also linked to high sample entropy for some LVA distributions, as a result of higher excitability promoting focal re-entrant sources. Lastly, $G_{Na}$ up-regulation increased the relative harmonic energy, indicating low complexity and a dominant role of the main atrial wave (Zeemering et al., 2018).

The ECG dominant frequency was mainly modulated by $I_{K1}$, with other ionic currents having a cumulative effect (Table 2). Besides $I_{K1}$ up-regulation, the general analysis identified a higher frequency in atria presenting increased $I_{to}$, $I_{NaK}$ and $I_{Na}$. Nevertheless, a thorough patient stratification demonstrated that atria with high $I_{to}$, $I_{NaK}$ and $I_{Na}$, but control $I_{K1}$, showed no significantly higher AF frequency than the overall population (Fig. 3*A*). This is illustrated through two simulated ECG recordings during AF (Fig. 3*B*), showing high and low frequency f-waves for $I_{K1}$ up- and down-regulation, respectively. Similarly, although faster AF episodes were obtained in atria that additionally presented low $I_{CaL}$, atria with increased $I_{CaL}$ showed significantly higher AF frequency than the overall population if $I_{to}$, $I_{K1}$, $I_{NaK}$ and $I_{Na}$ were up-regulated. As shown in Fig. 2*A*, the fastest AF episodes were obtained for the combination of increased $I_{to}$, $I_{K1}$, $I_{NaK}$, $I_{Na}$ and decreased $I_{CaL}$. This ionic current profile yielded the shortest tissue refractoriness ($108 \pm 10.3$ ms), which enabled a fast activation of re-entrant AF drivers.

## The ionic current dysregulation favouring AF maintenance is identified through ECG metrics, with $I_{K1}$ being the main determinant of AF dominant frequency

To assess whether the ionic density dysregulation presented in Fig. 2 could be identified non-invasively,

## High-frequency rotational activity in structurally-healthy areas coexist with rotors anchored around small and patchy LVA

Dominant frequency maps were computed in the 494 sustained AF episodes (>7 s) to analyze AF dynamics. Each AF episode exhibited a different range of AF frequencies. To enable comparisons on the location of

**Table 2. Effect of ionic current dysregulation on the ECG metrics during atrial fibrillation**

| % LVA left atrium | | Ionic current down($\downarrow$)- or up($\uparrow$)-regulation favouring an increase in | | | | |
|---|---|---|---|---|---|---|
| | | DF | OI | SE | SaE | RHE |
| Distribution 1 | 19.2 | $\uparrow G_{K1}(8)$, $\uparrow G_{NaK}(8)$, $\uparrow G_{to}(8)$, $\uparrow G_{Na}(8)$ | – | – | – | $\uparrow G_{Na}(7)$ |
| Distribution 2 | 20.8 | $\uparrow G_{K1}(8)$, $\uparrow G_{NaK}(8)$, $\uparrow G_{Na}(4)$ | – | – | $\uparrow G_{NaK}(3)$ | $\uparrow G_{Na}(7)$ |
| Distribution 3 | 22.2 | $\uparrow G_{K1}(8)$, $\uparrow G_{NaK}(8)$, $\uparrow G_{Na}(8)$, $\uparrow G_{to}(5)$ | – | $\downarrow G_{to}(4)$ | $\uparrow G_{NaK}(6)$ | $\uparrow G_{Na}(5)$ |
| Distribution 4 | 23.4 | $\uparrow G_{K1}(8)$, $\uparrow G_{NaK}(8)$, $\uparrow G_{to}(6)$, $\uparrow G_{Na}(6)$ | $\uparrow G_{to}(4)$ | $\uparrow G_{CaL}(3)$ | $\downarrow G_{CaL}(6)$ | $\uparrow G_{Na}(4)$ |
| Distribution 5 | 28.3 | $\uparrow G_{K1}(8)$, $\uparrow G_{NaK}(8)$, $\uparrow G_{to}(7)$ | – | $\downarrow G_{to}(3)$ | $\uparrow G_{K1}(5)$, $\uparrow G_{to}(5)$ | $\uparrow G_{Na}(4)$ |
| Distribution 6 | 39.9 | $\uparrow G_{K1}(8)$, $\uparrow G_{NaK}(8)$, $\uparrow G_{to}(8)$ | $\uparrow G_{CaL}(4)$ | $\uparrow G_{CaL}(6)$, $\uparrow G_{Kur}(4)$, $\downarrow G_{to}(3)$ | $\uparrow G_{to}(5)$, $\downarrow G_{CaL}(4)$ | $\uparrow G_{Na}(4)$ |
| Distribution 7 | 44.2 | $\uparrow G_{K1}(8)$, $\uparrow G_{NaK}(8)$, $\uparrow G_{Na}(8)$, $\uparrow G_{to}(7)$ | $\downarrow G_{NCX}(4)$ | – | – | $\uparrow G_{Na}(4)$ |
| Distribution 8 | 57.6 | $\uparrow G_{K1}(8)$, $\uparrow G_{to}(8)$, $\uparrow G_{NaK}(7)$, $\uparrow G_{Na}(6)$, $\downarrow G_{CaL}(5)$ | – | $\uparrow G_{CaL}(4)$ | $\downarrow G_{CaL}(5)$ | – |

Ionic current down- or up-regulation favoured a statistically-significant increase in the dominant frequency (DF), organization index (OI), spectral entropy (SE), sample entropy (SaE) and relative harmonic energy (RHE) compared to the baseline value. Statistical significance with $P < 0.001$ is indicated in bold; otherwise, $P < 0.05$. The number of ECG leads where the ionic current dysregulation increases each biomarker is shown in brackets next to the ionic density, with eight being the maximum. Data were analysed using the Wilcoxon rank sum test.

high-frequency regions across multiple AF episodes, every episode was represented by a discrete frequency map. The discrete map was developed by assigning to each cell of the atria one of three possible categories: very low activation frequency (frequencies below 10th percentile for the given AF episode), intermediate (10–90th percentile) and very high activation frequency (above 90th percentile). For each LVA distribution, density maps of high-frequency areas were constructed by superimposing high-frequency values obtained in multiple AF episodes. Figure 4*A* illustrates the transmembrane voltage of a representative AF episode and the resulting discrete frequency map. Figure 4*B* shows the bipolar voltage and density maps for different LVA distributions.

Rotors were not exclusive to the structurally-remodelled substrate. The down-regulation of $I_{CaL}$ and the up-regulation of $I_{K1}$, $I_{to}$ and $I_{NaK}$ contributed to the stabilization of functional rotors in healthy atrial tissue, appearing primarily in the right atrium (considered as non-structurally-remodelled in the present study). Importantly, rotational activity found in structurally-healthy areas coexisted with rotors anchored around LVA (Fig. 4*A*). Moreover, functional rotors exhibited a higher frequency (Fig. 4*A*). In frequency maps, these rotors appeared as high-frequency regions (i.e. arm of the re-entry), surrounding middle (i.e. border

of the core) and low-frequency regions (i.e. core of the re-entry). As described in previous studies (Bueno-Orovio et al., 2008; Heijman et al., 2014), the core of a functional re-entry is formed by excitable but unexcited tissue. This serves as validation of our discrete frequency maps, with the re-entrant cores consistently corresponding to low frequency regions for all considered LVA distributions.

Frequency density maps revealed that LVA were distinguished by low frequency areas (Fig. 4*B*). However, small and patchy LVA surrounded by healthy tissue (i.e. LVA distributions 2 and 5) showed high-frequency regions around LVA (Fig. 4*B*, blue arrows). The excitability gradient between LVA and healthy tissue created an arrhythmogenic substrate for rotor anchoring. When LVA were small, healthy tissue presenting high excitability, and short refractoriness (i.e. elevated $I_{K1}$ and reduced $I_{CaL}$) sustained rotational activity around LVA (between 2 and 3 s). Conversely, scattered and large LVA patches (LVA distribution 8) caused an excessive reduction of cellular excitability. Since the right atrium preserved high cellular excitability (i.e. modelled as non-LVA), high-frequency sources arising from the former prevented the anchoring of rotors in the structurally-remodelled left atrium. Thus, distributions with high percentage of LVA (LVA distribution 6–8) showed high-frequency regions only in the right atrium.

***In silico* drug trials with four reference compounds applied to 494 human atria provide a mechanistic explanation of anti-arrhythmic drug outcome and identify the inward currents ($I_{CaL}$ and $I_{Na}$) as critical for optimal stratification of AF patient to pharmacological treatment**

The 494 sustained AF episodes (>7 s) were subject to *in silico* trials with 12 pharmacological treatments (i.e. 6422 simulated AF episodes in total). Figure 5 compares the success rate obtained *in silico* and in human clinical trials (Fig. 5*A*), and the atrial ionic current substrate favouring pharmacological AF prevention (Fig. 5*B* and *C*).

Drug efficacy increased proportionally to a dose-dependent prolongation of refractoriness. This was less accentuated for vernakalant, with both doses achieving a similar efficacy despite a greater prolongation of refractoriness after vernakalant 30 $\mu$M ($+16.5 \pm 27.6$ *vs.* $+54.0 \pm 44.8$ ms with respect to control; vernakalant 10 *vs.* 30 $\mu$M). Conversely, a decrease in the refractory period derived from acute modelling of amiodarone ($-5.0 \pm 18.1$ and $-4.0 \pm 18.4$ ms; amiodarone 1.5 and 3

$\mu$M, respectively), which explained its very low efficacy (0.1–11%) (Fig. 5*A*). A prolongation of refractoriness ($+71.5 \pm 71.2$ ms compared to control), followed by an increase in prevention efficacy, resulted from applying chronic amiodarone modelling (i.e. 33% efficacy without $I_{CaL}$ blockade and 72–89% with progressive $I_{K1}$ inhibition). Thus, amiodarone showed the highest prevention efficacy within the tested treatments, consistent with clinical trials (Fig. 5*A*).

High success rates derived also from 70% $I_{Na}$ blockade (i.e. flecainide 2 $\mu$M) as a result of a great post-repolarization refractoriness. However, the prolongation of the refractory period, and thus flecainide prevention efficacy, dropped proportionally to a reduction in $I_{Na}$ blockade ($+171.5 \pm 164.9$ *vs.* $+100.0 \pm 131.4$ *vs.* $+18.0 \pm 35.2$ ms; prolongation of the refractory period after 70%, 60% and 40% $I_{Na}$ blockade). Similarly, the efficacy of digoxin was proportional to $I_{NaK}$ inhibition. Compared to 30% $I_{NaK}$ blockade, a 60% inhibition increased the refractory period by two-fold ($+22.0 \pm 35.0$ *vs.* $+53.0 \pm 57.7$ ms), as well as the efficacy of digoxin (60% $I_{NaK}$ inhibition is considered cardiotoxic and is

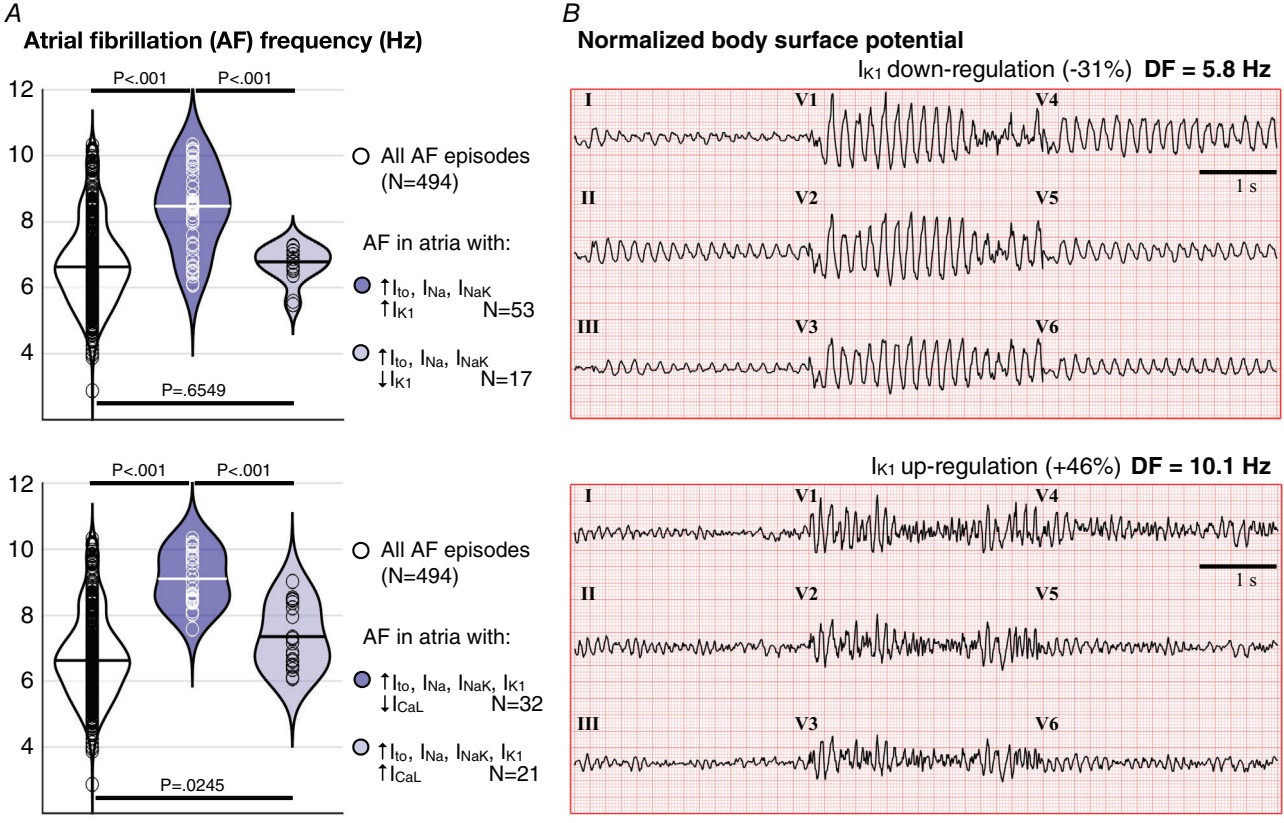

**Figure 3. Effect of $I_{K1}$ and $I_{CaL}$ on the ECG dominant frequency (DF)**
*A*, comparison of the atrial fibrillation (AF) dominant frequency in atria with $I_{to}$, $I_{NaK}$ and $I_{Na}$ up-regulation with *vs.* without $I_{K1}$ up-regulation (top), and with $I_{to}$, $I_{NaK}$, $I_{Na}$ and $I_{K1}$ up-regulation with *vs.* without $I_{CaL}$ down-regulation. *B*, simulated ECG recordings during AF in virtual atria with low and high $I_{K1}$. [Colour figure can be viewed at wileyonlinelibrary.com]

only calculated to show the main role of $I_{NaK}$ on AF perpetuation; see Discussion).

Interestingly, although pharmacological outcome was largely independent of the extent of structural remodelling, the atrial ionic current substrate determined treatment success. Figure 5*B* analyses the pharmacological outcome (i.e. AF termination *vs*. non-termination) of three drugs commonly used for rhythm control of AF according to the atrial ionic current profile (88 of the 100 profiles are included in the analysis since 12 failed to sustain AF) (Fig. 2). These ionic profiles have been grouped into three categories: profiles with a favourable drug response in all (100%) LVA distributions, in most of them (50–90% of LVA distributions) and in only a few (0–50%). Figure 5*C* infers potential patient stratification by differentiating the ionic current characteristics of atria responding to one drug or another.

Vernakalant lost efficacy in atria with significant $I_{CaL}$ down-regulation and $I_{NaK}$ up-regulation. As the major $I_{Kur}$ blocker considered, vernakalant prolonged the refractory period to a lesser extent in atria with low $I_{CaL}$ ($+80.0 \pm 36.4$ *vs*. $+15.8 \pm 26.3$ ms; prolongation of refractoriness in atria responding *vs*. non-responding to vernakalant). Because models refractory to vernakalant additionally showed high $I_{NaK}$, they were characterized by higher excitability (refractory period: $159.4 \pm 43.3$ *vs*. $124.2 \pm 17.5$ ms; models responding *vs*. non-responding to vernakalant), such that $I_{Kur}$ inhibition was not sufficient to terminate AF. Amiodarone was efficacious in most ionic current profiles (65/88) (Fig. 5*B*). However, it failed to prevent AF in virtual atria with the shortest refractoriness ($153.4 \pm 40.0$ *vs*. $118.3 \pm 16.9$ ms; atria responding *vs*. non-responding to amiodarone) because if significant $I_{K1}$ and $I_{NaK}$ up-regulation, and $I_{CaL}$

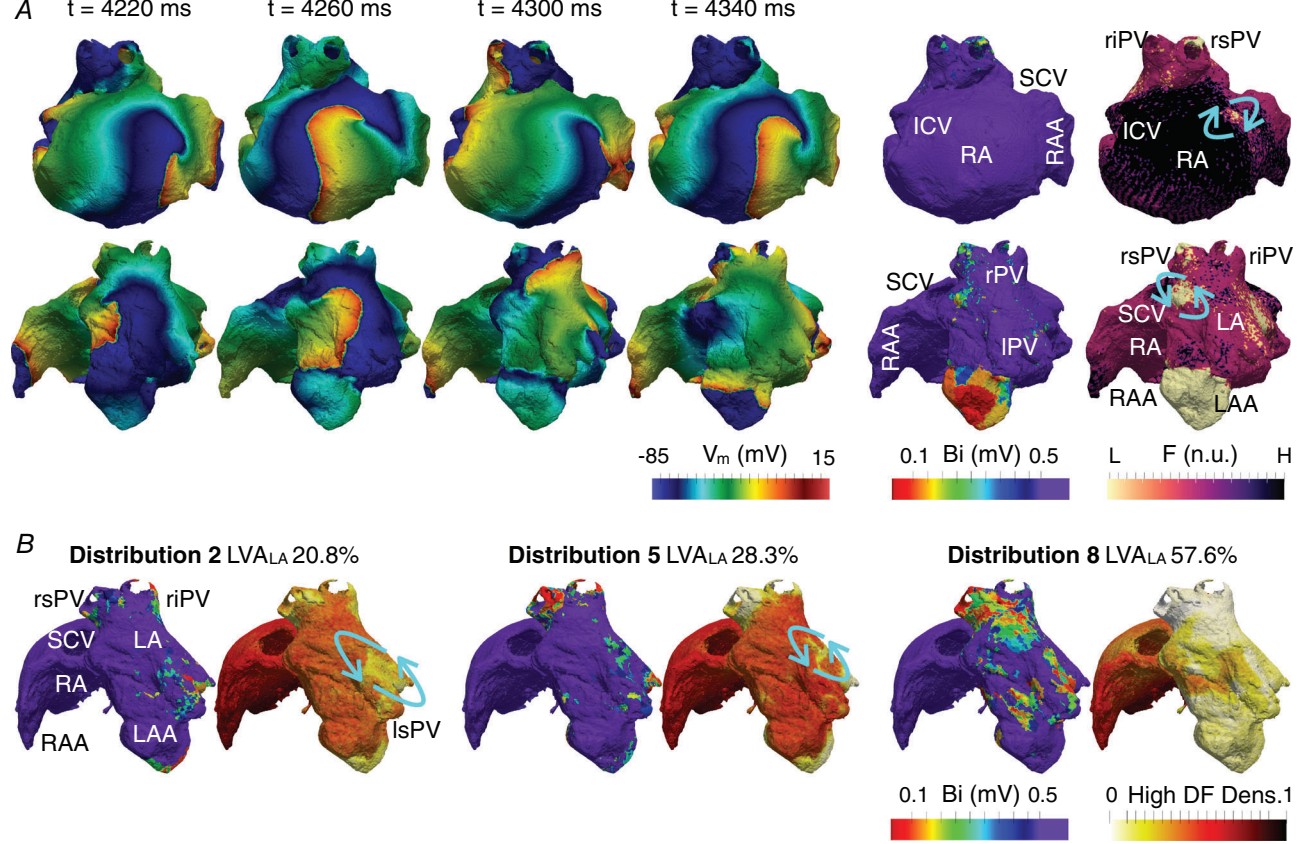

**Figure 4. Dominant frequency density maps for different low voltage areas (LVA) distributions**
*A*, consecutive snapshots of the transmembrane voltage ($V_m$) for a single AF episode. One rotor (top row) is located in the right atrium and one (bottom row) is anchored to a low voltage area of the left atrium (bipolar voltage, Bi < 0.5 mV). The discrete frequency (F, no units) map illustrates the core of the re-entry as a low (L) frequency region and the surroundings as middle and high (H) frequency regions. *B*, high dominant frequency (DF) density maps for three LVA distributions in the left atrium (LVA$_{LA}$): two with low one with high LVA density. The blue arrows highlight high-frequency regions adjacent to low density regions. Anatomical landmarks: RA-LA, right and left atrium; RAA-LAA, RA and LA appendage; SCV-ICV, superior and inferior cava vein; rPV-lPV, right and left pulmonary veins; rs-ri-ls-liPV, right superior, right inferior, left superior, left inferior pulmonary vein. [Colour figure can be viewed at wileyonlinelibrary.com]

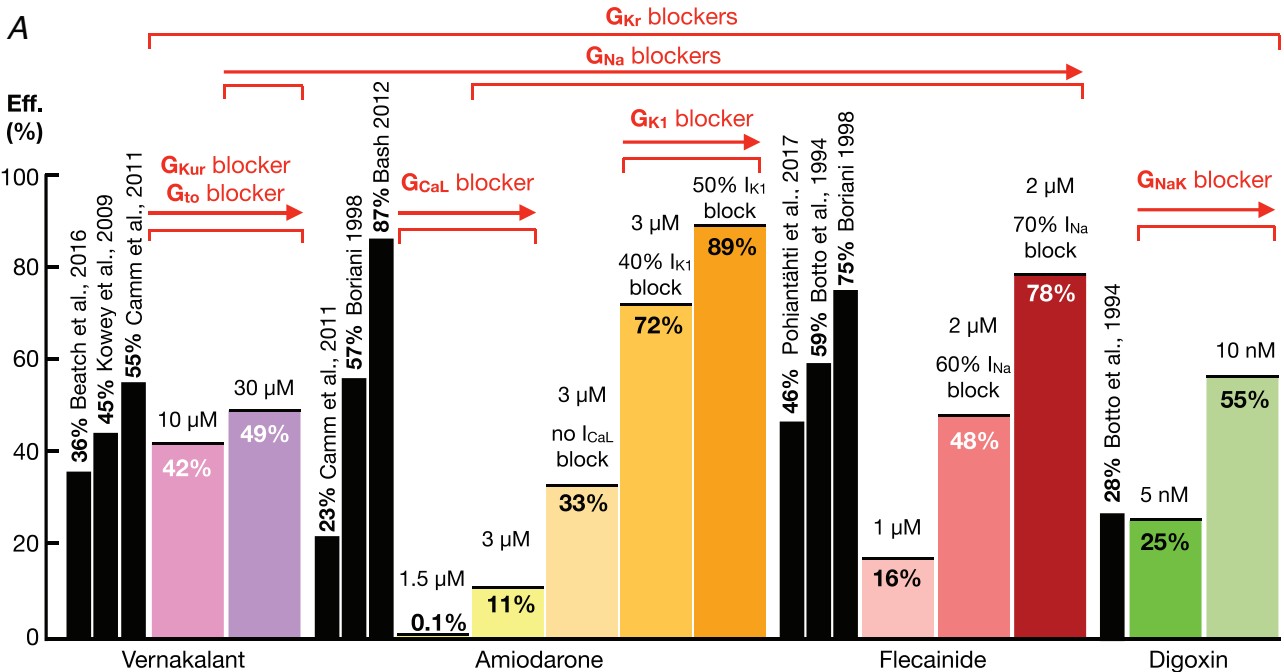

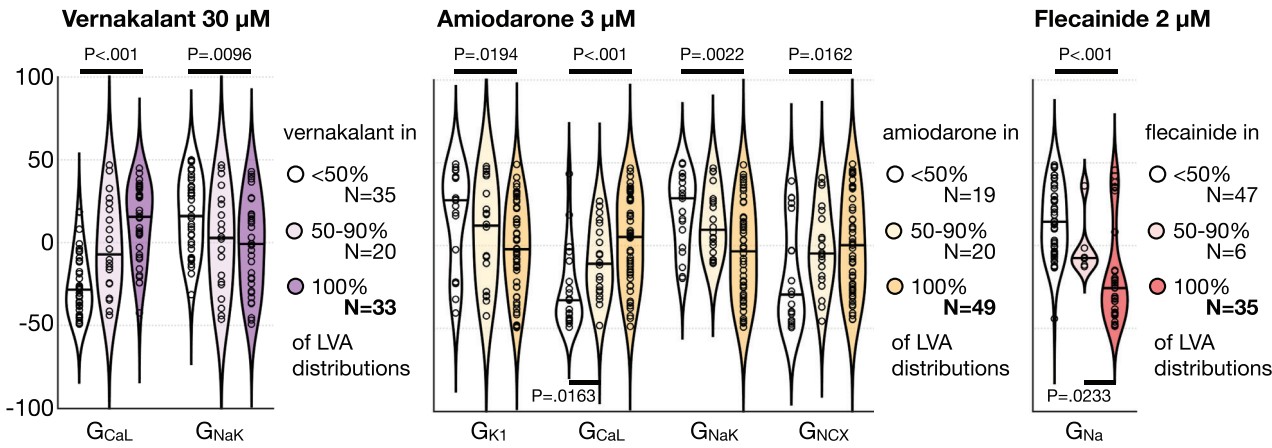

**B**

**Ionic current density variation (%) of ionic current profiles sustaining AF (N=88) and responding to:**

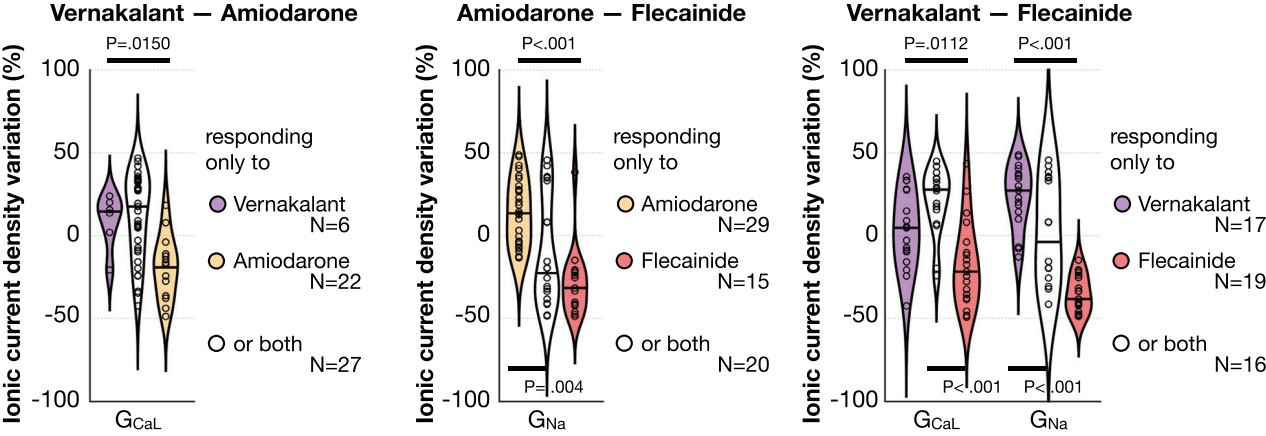

**C**

**From the ionic current profiles responding to Vernakalant (N=33), Amiodarone (N=49) and Flecainide (N=35)**

**Figure 5.** *In silico* trials over 494 AF episodes with 12 pharmacological treatments
*A*, comparison between the success rates (Efficacy, Eff.) obtained *in silico* (colour bars) and in human clinical trials (black bars). A threshold of 25% blockade at therapeutic plasma concentration has been considered to group drugs according to the ionic current channel they target. The arrows point towards greater ionic current density blockade. *B*, comparison of the ionic current density distribution of ionic current profiles responding and not responding to pharmacological treatment for a different number of LVA distributions. The three drugs considered are vernakalant 30 $\mu$M, amiodarone 3 $\mu$M without $I_{CaL}$ blockade and flecainide 2 $\mu$M with 60% $I_{Na}$ blockade. *C*, comparison of the ionic current densities of virtual patient models only responding to *drug-A* against those only responding to *drug-B* (where *drug-A* and *drug-B* are the three anti-arrhythmic drugs shown in *B*). Only ionic current densities presenting significant differences are displayed in the comparisons illustrated in *B* and *C*. [Colour figure can be viewed at wileyonlinelibrary.com]

and $I_{NCX}$ down-regulation. Finally, flecainide increased post-repolarization refractoriness through $I_{Na}$ inhibition. Therefore, virtual atria presenting $I_{Na}$ up-regulation were less sensitive to $I_{Na}$ blockade.

The comparison of the ionic current profiles of atria responding to either drug (Fig. 5C) revealed that vernakalant and amiodarone were superior to flecainide for AF termination when $I_{Na}$ was up-regulated. When $I_{CaL}$ and $I_{Na}$ were down-regulated, $I_{Na}$ inhibition was a very effective strategy (as explained above), such that flecainide was the desired approach for AF termination. When comparing amiodarone and vernakalant, vernakalant was favourable for $I_{CaL}$ up-regulation.

Thus, generally, flecainide was the preferred choice for inward current down-regulation. Conversely, for elevated $I_{Na}$, vernakalant and amiodarone were the better option; the former for high $I_{CaL}$ and the latter for atria presenting the highest cellular excitability ($I_{Na}$ up-regulation with additional $I_{CaL}$ down-regulation).

## Discussion

In the present study, *in silico* trials are conducted in a large cohort of human atria with electrophysiological and structural variability. A considerarion of electrophysiological variability is crucial for overcoming bias associated with drug effects because the success of pharmacological therapy depends on the specific ionic current substrate of the atria. Similarly, including structural variability is crucial to represent complex AF dynamics. As such, we have identified several structural and electrophysiological characteristics that modulate AF mechanisms and management, and have leveraged them to create a decision algorithm for guiding the decision-making of AF therapy. (i) Sustained AF ($>7$ s) develops in atria characterized by high excitability and short refractoriness, as a result of decreased $I_{CaL}$ and increased $I_{to}$, $I_{NaK}$ and $I_{K1}$. (ii) Progressive $I_{K1}$ up-regulation is required for AF maintenance as the extent of LVA increases in the left atrium. (iii) Large LVA infiltration in the left atrium yields absence of rotors in this chamber; conversely, regions of high atrial activity (i.e. high-frequency sources) appear adjacent to small and patchy LVA. (iv) Importantly, rotors anchored to structurally-remodelled areas coexist with functional rotors present in structurally-healthy tissue, with the latter exhibiting higher frequency. (v) *In silico* trials in 494 virtual patients with two to five doses of four anti-arrhythmic drugs (i.e. 6422 simulated AF episodes: 494 episodes under control conditions and 5928 after application of 12 treatments) identify the inward currents ($I_{CaL}$ and $I_{Na}$) as critical for optimal stratification of AF patient to pharmacological therapy. Flecainide is the most effective for $I_{Na}$ down-regulation. Conversely, when $I_{Na}$ is up-regulated, vernakalant and amiodarone are a better option, for high and low $I_{CaL}$, respectively. (vi) Notably, ECG markers could characterize this ionic current substrate and help with patient stratification.

### Successful pharmacological treatment is influenced by a patient's ionic current substrate

The ionic current substrate, rather than the extent, location or distribution of the structurally-remodelled substrate, played the most important role on pharmacological AF management. We previously demonstrated that the ionic current profile of the atria influences anti-arrhythmic drug outcome in structurally-healthy atria (Dasí et al., 2022). Moreover, several simulation studies (Margara et al., 2021; Passini et al., 2017, 2019, 2021) have indicated the relevance of ionic current variability to assess drug-induced myocardial proarrhythmia. This is important because it underlines the possibility of predicting the response to pharmacological treatment using a patient's individual ionic current profile (Capucci et al., 2018). Furthermore, as illustrated here and elsewhere (Dasi et al., 2022), the ECG holds potential for non-invasively characterizing this ionic substrate, which may improve patient stratification, cardiac safety and the efficacy of drug therapy. It is therefore highlighted the integral part of human-based *in silico* trials for gaining insights into cardiac pathology, response to pharmacological therapy (Rodriguez et al., 2016) and treatment personalization (Roney et al., 2022).

In our *in silico* trials, a success rate of 49% was observed after the virtual administration of vernakalant. We previously observed a success rate of 65% in structurally-healthy atria (Dasí et al., 2022). The latter

results are consistent with a randomized clinical trial conducted in patients with recent-onset AF (Simon et al., 2017), reporting an efficacy of 69% (34/49 patients). As in the present study, lower rates, such as 37% (81/221 patients) (Roy et al., 2008), 46% (59/129 patients) (Beatch & Mangal, 2016) and 52% (60/116 patients) (Camm et al., 2011), were obtained when additionally considering persistent AF patients, reflective of more advanced AF substrate.

We obtained low efficacies for acute modelling of amiodarone (i.e. 0.1–11%). It has been suggested that the high cardioversion of amiodarone results from the long-term accumulation of desethylamiodarone metabolite (Talajic, DeRoode & Nattel, 1987). Other studies advocate for amiodarone-induced $I_{Na}$ inhibition as the primary mechanism involving rotor termination (Wilhelms et al., 2013). When we modelled amiodarone without $I_{CaL}$ inhibition, a three-fold increase in its prevention efficacy (i.e. 33%) was observed. In a study conducted with rabbit ventricular cardiomyocytes, acute but not chronic exposure to amiodarone inhibited $I_{CaL}$ (Varró, Virág & Papp, 1996). Moreover, even when amiodarone was shown to be a strong $Ca^{2+}$ channel blocker experimentally (Nishimura, Follmer & Singer, 1989), it was suggested that very little blockade (i.e. 1–2%) would be expected under clinical conditions. Further experimental studies found that chronic treatment with amiodarone reduced $I_{NaK}$ by 33% (Grey et al., 1998) and exerted a strong inhibition of the inward rectifier $K^+$ currents (Guillemare et al., 2000). We found the inward rectifier $K^+$ channels to be crucial for AF maintenance (see below) and thus, when our formulation of chronic amiodarone considered progressive $I_{K1}$ blockade, success rates of 72–89% were observed, matching clinical trials (Bash et al., 2012).

Virtual flecainide administration also showed consistency between the simulated efficacy (i.e. 48–78%) and the success rate observed in clinical trials of 46% (46/100 patients) (Pohjantähti-Maaroos et al., 2019), 59% (20/34 patients) (Botto et al., 1994) and 75% (52/69) (Boriani et al., 1998). Different degrees of $I_{Na}$ inhibition were considered, including high degrees of blockade observed at fast activation rates (Moreno et al., 2011). Accordingly, we observed an increase in flecainide efficacy with increasing $I_{Na}$ blockade, particularly in atria with low $G_{Na}$. In patients with $I_{Na}$ down-regulation, additional $I_{Na}$ inhibition (i.e. flecainide) could pose a risk to the normal electrical conduction. However, the sever $I_{Na}$ blockade exerted by flecainide in the present study would only be expected at very high atrial rates (i.e. those observed during fibrillation). Under these high rates, flecainide application might even result in a complete loss of tissue excitability, as seen in atrial trabeculae after the application of compounds with a strong rate-dependent $I_{Na}$ blockade (Wettwer et al., 2013). Conversely, a minimal $I_{Na}$ blockade, and thus preserved conduction, would be expected during normal sinus rhythm (i.e. lower atrial rates).

A similar dose-dependent increase in the success rate derived from progressive $I_{NaK}$ inhibition (i.e. virtual digoxin administration). Digoxin is however administered at low doses because of the toxicity associated with cardiac glycosides (Bueno-Orovio et al., 2014). Nevertheless, although observational studies have reported an increased mortality with high plasma levels of digoxin (Hindricks et al., 2021), a recent randomized controlled trial reported a lower rate of adverse events with low-dose digoxin compared to bisoprolol (β-blocker) in patients with permanent AF and heart failure (Kotecha et al., 2020). In the present study, low-dose digoxin (i.e. 5 nm and 30% $I_{NaK}$ inhibition) had only a 25% efficacy, consistent with its current use as rate control, rather than rhythm control (Hindricks et al., 2021; Kotecha et al., 2020). However, digoxin-induced $I_{NaK}$ inhibition and the subsequent excitability reduction hold potential for terminating re-entrant drivers (Bueno-Orovio et al., 2014; Dasí et al., 2022), especially when combined with other anti-arrhythmics drugs.

## Fibrillation drivers are not exclusive of scarred areas

We identified small and patchy LVA as the preferred location for these re-entrant drivers. The predisposition of rotational activity for LVA has been widely demonstrated, in both clinical (Honarbakhsh et al., 2019) and simulation studies (Azzolin et al., 2022). This, together with the high prognostic value of LVA for predicting AF recurrence after catheter ablation (Masuda et al., 2018, 2020), led to the consideration of LVA ablation as an attractive personalized approach (Junarta et al., 2022). Nevertheless, recent clinical trials found no additional benefit from LVA ablation compared to pulmonary vein isolation alone (Masuda et al., 2020; Yang et al., 2022).

Furthermore, the presence of LVA might not necessarily correlate with AF drivers (Kircher et al., 2018). LVA show low specificity detecting AF sources, with only a small LVA proportion harboring proarrhythmic potential (Chen et al., 2019). Although transient rotational activity might localize within atrial regions showing complex fractionated electrograms (commonly seen in LVA), most regions with electrogram fractionation are not associated with rotational activity (Ghoraani et al., 2013). Similarly, in the present study we found that large and scattered LVA had a passive role on AF perpetuation, with high atrial activity arising only within small LVA.

This is in agreement with previous simulation studies (McDowell et al., 2015; Roney et al., 2016), reporting that large degrees of fibrosis prompted wavefront meandering and loss of anchors. However, as opposed to these studies,

we did not find spontaneous termination under these conditions because large LVA infiltration in the left atrium yielded rotor drift towards the right atrium. The latter, modelled as structurally-healthy tissue, always presented high atrial activity (i.e. high-frequency regions). Indeed, we commonly observed atrial activity of higher frequency in structurally-healthy regions, with functional rotors coexisting with rotors anchored to areas of scarring. This has been previously attributed to a hierarchical organization in activation frequency (Atienza et al., 2006), with high-frequency sources most likely harboring functional rotors (Rodrigo et al., 2014).

The coexistence of rotors in structurally-healthy and structurally-remodelled areas could explain the lack of beneficial impact of LVA ablation in paroxysmal AF patients. Masuda et al. (2020) suggested that the presence of LVA in these patients underlined a natural predisposition to develop arrhythmogenic substrate, which would promote AF even post-ablation. Further explanations considered the primary role of triggered ectopy compared to arrhythmogenic substrate on paroxysmal AF (Junarta et al., 2022; Masuda et al., 2020). In the present study, however, we present an additional hypothesis to the lack of efficacy of LVA ablation, based on the assumption that the structural substrate is not the only arrhythmogenic perpetuator. As illustrated in the present study, re-entrant sources arising in structurally-healthy regions may sustain AF, even when presented together with rotors anchored to structural heterogeneities.

### Ionic current dysregulation is crucial for AF maintenance

Several ionic current dysregulations contributed to the stabilization of the above-mentioned functional drivers, namely, the down-regulation of $I_{CaL}$ and the up-regulation of $I_{K1}$, $I_{to}$ and $I_{NaK}$. Indeed, atria presenting this ionic current substrate sustained AF irrespective of the LVA distribution. Numerous simulation studies (Boyle et al., 2019; Morgan et al., 2016; Zahid et al., 2016) have regarded fibrosis as the main determinant of AF inducibility, maintenance and overall dynamics. Conversely, in our population of virtual patient models with individual ionic current profiles, although LVA influenced AF dynamics (Fig. 4), the ionic current substrate played a crucial role. Clinically, AF inducibility and maintenance are common in human patients without structural heart disease (Kumar et al., 2012). Therefore, despite the central role of the structural remodelling on AF perpetuation, a more complex interaction between structural and functional perpetuators probably dictates AF dynamics (Heijman, Linz & Schotten, 2021).

As in our simulations, previous studies highlighted the role of increased inward rectifier K$^+$ channels on rotor stabilization and AF acceleration (Atienza et al., 2006; Pandit & Jalife, 2013). More specifically, decreased $I_{CaL}$ and increased $I_{K1}$ have been considered the primary electrical remodelling causing a progressive increase in the dominant frequency in paroxysmal AF (Martins et al., 2014). Similarly, we found that $I_{K1}$ up-regulation increased AF dominant frequency, independently of additional ionic current dysregulations. This increase was further exacerbated when combined with $I_{CaL}$ down-regulation, with both ionic current dysregulations yielding high-frequency f-waves in the ECG.

### Limitations and future studies

Drug action was simulated as simple pore-block models as the data required were available for the compounds investigated. However, neglecting the rate dependence of flecainide and vernakalant meant that all virtual patient models were subject to the same ionic current blockade. Because we observed very different AF dominant frequencies, future studies should attempt to take the information of the AF frequency into account to simulate accurately the effects of compounds with rate dependence.

Furthermore, the patient-specific modelling of LVA only considered structurally-remodelled left atrium. Many studies have reported that the extent of LVA in the left atrium has prognostic value for predicting AF recurrence after pulmonary vein isolation (Masuda et al., 2020). Thus, we aimed to isolate the role of left atrial LVA on pharmacological treatment success. Nevertheless, the findings may be determined by the relative distribution of LVA between the atria, such that future studies should consider accurate LVA infiltration in both the right and left atrium. Similarly, although we considered different LVA distributions in the left atrium, all simulations were performed with the same atrial anatomy. The structural remodelling of the atria is also associated with variations in atrial shape and size, which should be accounted in future studies.

Virtual atria included regional electrophysiological heterogeneities. However, the pulmonary veins and the left atrial posterior wall shared the same electrical properties, and the anatomical insertions between the two were not included in the atrial models. Although modelling these features might have slightly changed AF dynamics, this work focused on identifying how LVA infiltration in the left atrial body influenced AF mechanisms and pharmacological treatment. Because ectopic initiation at the pulmonary veins was not simulated, an accurate representation of the latter was beyond the scope of the present study.

The AF induction protocol imposed spiral wave re-entries as the initial condition for the simulation. Thus, slight changes in the number and location of these phase

singularities might influence AF dynamics. However, we observed that the mechanisms maintaining AF were primarily modulated by the structural and ionic current substrate. As shown here and elsewhere (Dasi et al., 2022; Matene & Jacquemet, 2012), atria with high $I_{CaL}$ presented high propensity for APD alternans and wavefront fractionation. Under this ionic current substrate, Matene & Jacquemet (2012) observed that the complexity of AF dynamics was not controlled by the initial number of phase singularities. On the other extreme, rotors tended to anchor in structurally-remodelled substrate (i.e. LVA) for atria with low $I_{CaL}$. Similarly, in this case, the initial position of spiral waves did not modulate the resulting AF dynamics. Most importantly, the bias associated with the exact selection of phase singularities was addressed by the comparative nature of the present study. This is, the method chosen for inducing AF ensured that all scenarios tested shared comparable initiation conditions (Roney et al., 2022).

Finally, we have shown here and elsewhere (Dasi et al., 2022) that the ECG holds potential for non-invasively characterizing the ionic current substrate of the atria. Nevertheless, the simulated recordings were absent of noise, the QRS was already abstracted (no need for QRS cancellation) and we had perfect control over the ionic current densities of virtual atria. The clinical translation would therefore require a rigorous analysis with human ECGs to validate the simulated results.

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

## Additional information

### Data availability statement

The original contributions presented in the study are included in the article, further inquiries can be directed to the corresponding authors.

### Competing interests

The authors declare that they have no competing interests.

### Author contributions

All authors equally contributed to the conception of the work, revising it critically for important intellectual content, and gave their final approval of the version to be published, ensuring that questions related to the accuracy or integrity of any part of the work were appropriately investigated and resolved. A.D. was responsible for conducting all the simulations, analysis of results and drafting the work. B.R. and A.B.-O. acted as supervisors of the research. M.T.B.P., R.S.W., T.R.B. and R.S. mainly contributed to methodological aspects.

### Funding

This work was supported by a British Heart Foundation (BHF) Intermediate Fellowship (FS/17/22/32 644 to AB-O), a Wellcome Trust Senior Fellowship in Basic Biomedical Sciences (214 290/Z/18/Z to BR), the CompBioMed Centre of Excellence in Computational Biomedicine (European Union's Horizon 2020; grant agreements 675 451, 823 712) and the Oxford Biomedical Research Centre. This project has also received funding from the European Union's Horizon 2020 research and innovation programme under the Marie Skłodowska-Curie grant agreement No.860974 (to AD). We acknowledge additional support from an Infrastructure for Impact Award from the National Centre for the Replacement, Refinement and Reduction of Animals in Research (NC/P001076/1), the Oxford BHF Centre of Research Excellence (RE/13/1/30 181), PRACE, Piz Daint at the Swiss National Supercomputing Centre, Switzerland (ICEI-PRACE grants icp005 and icp013) and Fapemig.

For the purpose of Open Access, the authors have applied a CC BY public copyright license to any Author Accepted Manuscript (AAM) version arising from this submission.

### Keywords

atrial fibrillation, *in silico* drug trials, ionic currents, low voltage areas

### Supporting information

Additional supporting information can be found online in the Supporting Information section at the end of the HTML view of the article. Supporting information files available:

**Statistical Summary Document**
**Peer Review History**

