## [Peer Review History · The Journal of Physiology]

What determines the optimal pharmacological treatment of atrial fibrillation? Insights from in-silico trials in 800 virtual atria

Albert Dasí, Michael T.B. Pope, Rohan S. Wijesurendra, Tim R. Betts, Rafael Sachetto, Alfonso Bueno Orovio, and Blanca Rodriguez

DOI: 10.1113/JP284730

Corresponding author(s): Albert Dasí (albert.dasiimartinez@cs.ox.ac.uk)

The following individual(s) involved in review of this submission have agreed to reveal their identity: Trine Krogh-Madsen (Referee #1)

Review Timeline:

Submission Date:	22-Mar-2023
Editorial Decision:	18-Apr-2023
Revision Received:	07-Jun-2023
Editorial Decision:	23-Jun-2023
Revision Received:	03-Jul-2023
Accepted:	05-Jul-2023

Senior Editor: Harold Schultz

Reviewing Editor: Eleonora Grandi

Transaction Report:

Dear Dr Dasi,

Re: JP-RP-2023-284730 "What determines the optimal pharmacological treatment of atrial fibrillation? Insights from in-silico trials in 800 virtual atria" by Albert Dasi, Michael T.B. Pope, Rohan S. Wijesurendra, Tim R. Betts, Rafael Sachetto, Alfonso Bueno Orovio, and Blanca Rodriguez

Thank you for submitting your manuscript to The Journal of Physiology. It has been assessed by a Reviewing Editor and by 2 expert referees and we are pleased to tell you that it is potentially acceptable for publication following satisfactory major revision.

LANGUAGE EDITING AND SUPPORT FOR PUBLICATION: If you would like help with English language editing, or other article preparation support, Wiley Editing Services offers expert help, including English Language Editing, as well as translation, manuscript formatting, and figure formatting at www.wileyauthors.com/eoo/preparation. You can also find resources for Preparing Your Article for general guidance about writing and preparing your manuscript at www.wileyauthors.com/eoo/prepresources.

REVISION CHECKLIST:

We look forward to receiving your revised submission.

Yours sincerely,

Harold D Schultz
Senior Editor
The Journal of Physiology
<https://jp.msubmit.net>
<http://jp.physoc.org>
The Physiological Society
Hodgkin Huxley House
30 Farringdon Lane
London, EC1R 3AW
UK
<http://www.physoc.org>
<http://journals.physoc.org>

REQUIRED ITEMS

-Author photo and profile. First (or joint first) authors are asked to provide a short biography (no more than 100 words for one author or 150 words in total for joint first authors) and a portrait photograph. These should be uploaded and clearly labelled with the revised version of the manuscript. See Information for Authors for further details.

-The contact information provided for the person responsible for 'Research Governance' at your institution is an author on this paper. Please provide an alternative contact who is not an author on this paper or confirm that the author whose email was provided has sole responsibility for research governance. This is the person who is responsible for regulations, principles and standards of good practice in research carried out at the institution, for instance the ethical treatment of animals, the keeping of proper experimental records or the reporting of results.

-Please upload separate high-quality figure files via the submission form.

-Please ensure that any tables are in Word format and are, wherever possible, embedded in the article file itself.

-A Statistical Summary Document, summarising the statistics presented in the manuscript, is required upon revision. It must be on the Journal's template, which can be downloaded from the link in the Statistical Summary Document section here: https://jp.msubmit.net/cgi-bin/main.plex?form_type=display_requirements#statistics

-Papers must comply with the Statistics Policy https://jp.msubmit.net/cgi-bin/main.plex?form_type=display_requirements#statistics

In summary:

-If n {less than or equal to} 30, all data points must be plotted in the figure in a way that reveals their range and distribution. A bar graph with data points overlaid, a box and whisker plot or a violin plot (preferably with data points included) are acceptable formats.

-If $n > 30$, then the entire raw dataset must be made available either as supporting information, or hosted on a not-for-profit repository e.g. FigShare, with access details provided in the manuscript.

- 'n' clearly defined (e.g. x cells from y slices in z animals) in the Methods. Authors should be mindful of pseudoreplication.

- All relevant 'n' values must be clearly stated in the main text, figures and tables, and the Statistical Summary Document (required upon revision)

- The most appropriate summary statistic (e.g. mean or median and standard deviation) must be used. Standard Error of the Mean (SEM) alone is not permitted.

- Exact p values must be stated. Authors must not use 'greater than' or 'less than'. Exact p values must be stated to three significant figures even when 'no statistical significance' is claimed.

- Statistics Summary Document completed appropriately upon revision

- A Data Availability Statement is required for all papers reporting original data. This must be in the Additional Information section of the manuscript itself. It must have the paragraph heading "Data Availability Statement". All data supporting the results in the paper must be either: in the paper itself; uploaded as Supporting Information for Online Publication; or archived in an appropriate public repository. The statement needs to describe the availability or the absence of shared data. Authors must include in their Statement: a link to the repository they have used, or a statement that it is available as Supporting Information; reference the data in the appropriate section(s) of their manuscript; and cite the data they have shared in the References section. Whenever possible the scripts and other artefacts used to generate the analyses presented in the paper should also be publicly archived. If sharing data compromises ethical standards or legal requirements then authors are not expected to share it, but must note this in their Statement. For more information, see our Statistics Policy.

- Please include an Abstract Figure file, as well as the figure legend text within the main article file. The Abstract Figure is a piece of artwork designed to give readers an immediate understanding of the research and should summarise the main conclusions. If possible, the image should be easily 'readable' from left to right or top to bottom. It should show the physiological relevance of the manuscript so readers can assess the importance and content of its findings. Abstract Figures should not merely recapitulate other figures in the manuscript. Please try to keep the diagram as simple as possible and without superfluous information that may distract from the main conclusion(s). Abstract Figures must be provided by authors no later than the revised manuscript stage and should be uploaded as a separate file during online submission labelled as File Type 'Abstract Figure'. Please ensure that you include the figure legend in the main article file. All Abstract Figures should be created using BioRender. Authors should use The Journal's premium BioRender account to export high-resolution images. Details on how to use and access the premium account are included as part of this email.

- Please include a full title page as part of your article (Word) file (containing title, authors, affiliations, corresponding author name and contact details, keywords, and running title).

EDITOR COMMENTS

Reviewing Editor:

The manuscript is an important step forward in simulating ionic and structural heterogeneity in atrial fibrillation. Both reviewers identified several strengths, but also noted several issues that require substantial revision and clarification. The reviewing editor concurs.

Senior Editor:

The manuscript has been reviewed and given sufficient merit to move forward in order to address major issues in the paper. Major concerns were that more detail is needed in the methodological descriptions, a much better explanation of the material in Figure 2 is needed in addition to the fact that the data shown in the figure is not accurate and may impact Figure 3, concerns regarding the interpretation of Figure 3, anatomical markers needed in Figure 4, concerns regarding the interpretation of Figure 5, and concerns regarding text in the manuscript.

Please review the statistical reporting policy for the journal. Actual P values must be stated in text, figures and tables unless $P < .001$. If there are questions, please contact the journal.

Please be aware that revision of the manuscript does not at this point guarantee acceptance for publication.

REFeree COMMENTS

Referee #1:

In this manuscript, Dasi et al. presents the results from simulations employing ionic current heterogeneity coupled with structural heterogeneity (low voltage areas) in the human atria. The authors generate a population of 100 atrial cell ionic models that they combine with 8 whole-atria models of different low voltage area distributions to generate 800 virtual atria. They then simulate the efficacy of several anti-arrhythmic drugs in terminating simulated atrial fibrillation and correlate drug efficacy with ionic properties. In this way, the manuscript represents an important extension of population heterogeneity (usually modeled at the cellular level) into tissue-level models for a condition and system (atrial fibrillation) where both ionic and structural heterogeneity is prevalent.

I have a few comments/questions:

Simulated AF is assumed sustained after 7 s. Could the authors show the time course of AF maintenance over the first 7 s to demonstrate that 7 s is likely sufficient time? This is important because drug simulations are done subsequently and conversion of AF is considered to be caused by the drug.

2. In Fig. 2B: Was there a pattern across individual currents such that it was always, say, the same 1-4 LVA distributions that were in the light gray bars, the same additional LVA distributions in the mid-gray bars (4-6), and the same final distributions additional for the dark gray bars? Could this information be included in the figure? (Also the 1-4 and 4-6 categories overlap as both have 4, please correct)

3. Please add how ventricular electrical signals were removed from the ECG (Fig 2A).

4. Table 2 is a bit dense to read. Maybe color each conductance in a separate font color

5. Fig. 3: I'm confused about the comparisons. Why not compare ionic current up-regulation with control IK1 (black) to ionic current up-regulation (light gray) instead of ionic current down-regulation (white)? And similarly compare ionic current up-regulation + ICaL down-regulation (dark gray) to ionic current up-regulation (light gray) instead of ionic current down-regulation (white)? Also, it's not clear what "control IK1" means. I thought it would be an increase to compensate for low GK1 in the LVA but that doesn't seem to work in the right direction. Please clarify. Finally, the atria are too small to see - maybe the torso is not needed here as it was in a previous figure.

6. Fig. 4: Please add some anatomical markers to help orient the reader. Please define "discrete frequency" and explain why it is low at the core.

7. For the amiodarone simulations, the most significant non-terminating ionic properties seem to be low GCaL and low GNCX but I don't think these are mentioned.

8. Flecainide is efficient for AF termination when GNa is reduced. Is there a concern of compromising normal conduction with flecainide to a patient with low GNa?

9. Fig 5: I'm having a hard time connecting the AF terminated vs not terminated ratios (e.g., 50 vs 38 for vernakalant) in panel B to the success rates in panel A. I'm assuming it has to do with the LVA levels but please provide info/clarification on this.

Referee #2:

Major comments:

1. Could the reviewers please provide more information on their implementation of imposing spiral wave reentry (Matene and Jacquemet 2012)? Beyond citing the original paper it is important that readers understand how this was done in the current context. What were the initial conditions (e.g., how many phase singularities, direction of rotation, distance between phase singularities)? Adding this information is extremely important for reproducibility.
2. Furthermore, related to the above point #1, please discuss the sensitivity of the rotor imposition methodology to aforementioned initial conditions? Is it possible that slight changes in initial phase singularity position might render a case inducible with AF when it was previously non-inducible (or vice versa?)
3. In Figure 2A, the authors show simulated ECG recordings and clinical ECG recordings. Although this figure is adapted from a prior publication, several clarifications and changes are needed. First, in this context, it is unclear exactly how the atrial-only clinical ECG recordings were derived. Are these invasive measurements or non-invasive measurements? What type of filtering was used to achieve this schematic? Second, what are the respective time and voltage scales for both simulation and clinical signals? At minimum, these should be added. Third, after examining the original version of the figure from Lankveld et al. side-by-side with the version used in this paper, it appears the authors accidentally flipped the time series data about the horizontal axis (i.e., the signals are left-right mirrored). This leads to the final point, which is that visual analysis suggests that there is not a solid relationship between the simulated and clinical ECGs. The way the data are presented might lead a naïve reader to assume this is some kind of quantitative validation, when the point (at best) seems to be that the signals are qualitatively similar-ish. The usefulness of the comparison is somewhat undermined by the fact that the authors accidentally flipped the signals from the original paper and the data still look basically "similar".
4. Further to point #3 above, this may also prompt a rethinking regarding the usefulness of presenting simulated ECG data in Figure 3.
5. After all the ionic changes made (population model [variability {plus minus}50%]; regional ionic differences; reduction in ion channel conductance for LVA regions; and drug associated changes for Figure 5) is it possible that some of these models were deranged into a non-physiologic state, and thus failed to reliably conduct altogether (i.e., following imposition of spiral waves no further excitation occurs)? In other words, could the authors please provide some validation that following ionic parameter changes the resulting functionality at the organ scale remains physiologically plausible?
6. Lines 495-515: Increased LVA infiltration in the left atria leading to rotors in the structurally-healthy right atria coexisting with rotors in areas of structural-remodeling is an interesting finding. Could it be possible that these right atrial rotors exist due to an imposed spiral wave in that region? In this case, a reentrant source arises from the induction method rather than any physiologic manner (i.e., triggered ectopy or downregulation of ICaL and upregulation of IK1, Ito and INaK as the authors describe).

Minor comments:

Lines 152-154: What adjustments were made to account for the regional ionic differences, and where did the information come from?

Figure 2B: What was the rationale for grouping in an uneven manner? The manuscript would be strengthened if there were a clearly defined basis for determining the number of LVA distributions in each group.

Line 30: Clarify sentence to "pharmacological" or "drug-based" treatment.

Figure 3: Is there a reason boxplots were used instead of violin plots?

Lines 256-267: The authors should consider qualifying the language in this paragraph as changes in ionic current dysregulation were not reliably identifiable for all distributions of LVA. For example, in low LVA burden cases G_{CaL} and G_{to} did not appear to affect spectral entropy with statistical significance. Similar cases can be made for G_{NaK}/sample entropy and G_{Na}/relative harmonic energy.

END OF COMMENTS

Confidential Review

22-Mar-2023

EDITOR COMMENTS

Reviewing Editor:

The manuscript is an important step forward in simulating ionic and structural heterogeneity in atrial fibrillation. Both reviewers identified several strengths, but also noted several issues that require substantial revision and clarification. The reviewing editor concurs.

Senior Editor:

The manuscript has been reviewed and given sufficient merit to move forward in order to address major issues in the paper. Major concerns were that more detail is needed in the methodological descriptions, a much better explanation of the material in Figure 2 is needed in addition to the fact that the data shown in the figure is not accurate and may impact Figure 3, concerns regarding the interpretation of Figure 3, anatomical markers needed in Figure 4, concerns regarding the interpretation of Figure 5, and concerns regarding text in the manuscript.

We thank the editor and the reviewers for the very important feedback.

Considerable improvements are made in Figure 2-5 and their explanations following the suggestions above.

Please review the statistical reporting policy for the journal. Actual P values must be stated in text, figures and tables unless $P < .001$. If there are questions, please contact the journal.

The figures and the text have been modified to comply with the statistical reporting policy of the journal.

However, simulation studies go beyond the use of statistical tests and include a comprehensive evaluation of the physiological significance of the findings, regardless of specific P values. This represents the advantage of using populations of models calibrated and validated with experimental and clinical data for gaining mechanistic insights into cardiovascular variability, in that the effective size obtained from simulated data can be used as estimates for power calculations.

Please be aware that revision of the manuscript does not at this point guarantee acceptance for publication.

REFEREE COMMENTS

Referee #1:

In this manuscript, Dasi et al. presents the results from simulations employing ionic current heterogeneity coupled with structural heterogeneity (low voltage areas) in the human atria. The authors generate a population of 100 atrial cell ionic models that they combine with 8 whole-atria models of different low voltage area distributions to generate 800 virtual atria. They then simulate the efficacy of several anti-arrhythmic drugs in terminating simulated atrial fibrillation and correlate drug efficacy with ionic properties. In this way, the manuscript represents an important extension of population heterogeneity (usually modeled at the cellular level) into tissue-level models for a condition and system (atrial fibrillation) where both ionic and structural heterogeneity is prevalent.

I have a few comments/questions:

1. **Simulated AF is assumed sustained after 7 s. Could the authors show the time course of AF maintenance over the first 7 s to demonstrate that 7 s is likely sufficient time? This is important because drug simulations are done subsequently and conversion of AF is considered to be caused by the drug.**

We thank the reviewer for their comment and we have rewritten the relevant description to avoid misunderstandings. Drug simulations were not applied subsequent to the 7 s but during the same first 7 s than the control AF episode to ensure that drug action was the real cause of AF termination. We have provided a clearer explanation on this point:

Lines 198-202: *Sustained (>7s) AF episodes were subjected to the virtual administration of four drugs commonly used for AF treatment (Hindricks et al., 2021). Their administration was modeled 2 seconds from AF initiation and the episode was continued until completion of the original 7 s (i.e., 2 s in the absence of intervention and 5 s after virtual drug administration, Matene and Jacquemet, 2012).*

We conducted 7 s simulations as in *Matene and Jacquemet, 2012*. The time course of AF during these 7 s is illustrated in Figure 2 (ECG) and in Figure 4 (snapshots of the transmembrane voltage). We can provide additional media (i.e., videos of representative AF episodes) as necessary.

2. **In Fig. 2B: Was there a pattern across individual currents such that it was always, say, the same 1-4 LVA distributions that were in the light gray bars, the same additional LVA distributions in the mid-gray bars (4-6), and the same final distributions additional for the dark gray bars? Could this information be included in the figure? (Also, si the 1-4 and 4-6 categories overlap as both have 4, please correct).**

We thank the reviewer for their comment. To reveal this information, a new grouping of the ionic current profiles has been implemented, and the patterns between ionic current profiles and LVA distributions have been investigated.

This has been added in the main manuscript:

Lines 234-240. *Figure 2B characterizes the atrial ionic current substrate favoring AF maintenance. The 100 ionic current profiles included in this study have been grouped into three categories according to their proarrhythmic potential: profiles sustaining AF (>7s) for less than two LVA distributions (less than 25% of the LVA distributions studied), between two and six (25-75% of the LVA distributions) and for more than six LVA distributions (more than 75% of the LVA distributions). Figure 2C and 2D show the left atrium parcellation and the ionic current substrate needed for AF maintenance as the LVA extension increased.*

Lines 251 and 255-260. *IK1 up-regulation was especially needed as the LVA density increased in the left atrium [...] This is illustrated in the 32 ionic current profiles yielding AF maintenance in 2-6 LVA distributions. From the 32 profiles, 9 sustained AF (>7s) primarily in smaller LVA extensions (20-30% LVA), 9 in larger (40-60% LVA) and 14 in both. Compared to ionic current profiles sustaining AF in smaller LVA, those profiles engaging with severe structural remodeling showed a significantly higher GK1. This difference was accentuated when considering the LVA extension in the left atrial posterior wall (Figure 2C, regions 1-4).*

3. Please add how ventricular electrical signals were removed from the ECG (Fig 2A).

We appreciate the reviewer's comment. The clinical ECG is adapted from **Lankveld et al., 2016**, and a detailed explanation of QRS removal is provided in the original publication. A brief summary is added in the figure legend:

Lines 264-267: *The clinical ECG was recorded on the day of electrical cardioversion with a sampling frequency of 250 Hz. Postprocessing consisted of a zero-phase band-pass filter between 1 and 100 Hz, a 50-Hz notch filter and cancellation of ventricular signals: QRST-waves. For further details refer to Lankveld et al., 2016.*

In our simulations, we only considered the atrial anatomy. Thus, the ventricular electrical activity, and the resulting signals, were not simulated.

This has been explained in the methods section:

Lines 186-187: *Simulated 8-lead (Leads I, II, V1-V6) ECGs were computed for sustained (>7s) AF episodes as in Gima and Rudy, 2002. Only atrial electrical signals were simulated, so that ventricular cancellation (QRS- and T-wave removal) was not needed.*

4. Table 2 is a bit dense to read. Maybe color each conductance in a separate font color

We thank the reviewer for their comment. Their suggestions have been included in Table 2.

5. Fig. 3: I'm confused about the comparisons.

- a. **Why not compare Ionic current up-regulation with control IK1 (black) to Ionic current up-regulation (light gray) instead of Ionic current down-regulation (white)? And similarly compare Ionic current up-regulation + ICaL down-regulation (dark gray) to Ionic current up-regulation (light gray) instead of Ionic current down-regulation (white)? Also, it's not clear what "control IK1"**

means. I thought it would be an increase to compensate for low GK1 in the LVA but that doesn't seem to work in the right direction. Please clarify.

A new comparison according to the reviewer suggestions has been considered. The text and **Figure 3** have been modified accordingly:

Lines 296-304: *The ECG dominant frequency was mainly modulated by GK1 and GCaL. In the general analysis (Table 2) higher AF dominant frequency was observed in atria presenting Ito, IK1, INaK, INa up-regulation and ICaL down-regulation. Nevertheless, a thorough patient stratification demonstrated that atria with high Ito, INaK and INa but control IK1, showed no significantly higher AF frequency than the overall population (Figure 3A). This is illustrated through two simulated ECG recordings during AF (Figure 3B), showing high and low frequency f-waves for IK1 up- and down-regulation, respectively. Similarly, while the fastest AF episodes were obtained in atria with additionally low ICaL, atria with increased ICaL still presented significantly higher AF frequency than the overall population if Ito, IK1, INaK, INa were up-regulated.*

- b. Finally, the atria are too small to see - maybe the torso is not needed here as it was in a previous figure.**

We acknowledge that neither the torso nor the atria were adding useful information to the figure and thus have been removed.

- 6. Fig. 4: Please add some anatomical markers to help orient the reader. Please define "discrete frequency" and explain why it is low at the core.**

We thank the reviewer for their comment. Different AF episodes exhibited different ranges of AF frequencies. In order to compare the location of high-frequency regions across multiple AF episodes, every episode was represented by a discrete frequency map. The discrete map was developed assigning each cell of the atrial anatomy to one of three possible categories: cells with very low activation frequency (frequencies of that AF episode below 10th percentile), cells with intermediate (10-90th percentile) and cells with very high activation frequency (above 90th percentile).

This has been re-written in the text:

Lines 312-318: *Dominant frequency maps were computed in the 494 sustained AF episodes (>7s) to analyze AF dynamics. Each AF episode exhibited a different range of AF frequencies. To enable comparisons on the location of high-frequency regions across multiple AF episodes, every episode was represented by a discrete frequency map. The discrete map was developed by assigning to each cell of the atria one of three possible categories: very low activation frequency (frequencies below 10th percentile for the given AF episode), intermediate (10-90th percentile) and very high activation frequency (above 90th percentile).*

The anatomical landmarks are also added in Figure 4, and the explanation of why the frequency is low at the core is provided in the text:

Lines 352-354: *Anatomical landmarks. RA-LA: Right and left atrium; RAA-LAA: RA and LA appendage; SCV-ICV: Superior and inferior cava vein. rPV-IPV: Right and left pulmonary veins. rs-ri-ls-liPV: Right superior, right inferior, left superior, left inferior pulmonary vein.*

Lines 330-333: *As described in previous studies (Heijman et al., 2014; Bueno-Orovio et al., 2008), the core of a functional re-entry is formed by excitable but unexcited tissue. This serves as validation of our discrete frequency maps, with the re-entrant cores consistently corresponding to low frequency regions for all considered LVA distributions.*

7. For the amiodarone simulations, the most significant non-terminating ionic properties seem to be low G_{CaL} and low G_{NCX} but I don't think these are mentioned.

We thank the reviewer for their comment. The influence of G_{CaL} and G_{NCX} is now added in the main text.

Lines 395-398: *Amiodarone was efficacious in most ionic current profiles (65/88, Figure 5B). However, it failed to prevent AF in virtual atria with the shortest refractoriness (153.4±40.0 vs. 118.3±16.9 ms; atria responding vs. non-responding to amiodarone), due to significant IK1 and INaK up-regulation, and ICaL and INCX down-regulation.*

8. Flecainide is efficient for AF termination when G_{Na} is reduced. Is there a concern of compromising normal conduction with flecainide to a patient with low G_{Na}?

We thank the reviewer for the very interesting observation. Severely blocking G_{Na} in a patient with already G_{Na} down-regulation poses an undeniable risk for the normal conduction. However, the high G_{Na} block (60-70%) considered in this study tries to mimic the strong rate-dependent effects of Class Ic blockers. This block would not be expected during normal sinus rhythm, but instead, during episodes of atrial fibrillation. Therefore, under slow activation rates, the risk of compromising the normal conduction of the heart would be greatly reduced.

This has been added in the main text:

Lines 210-211: *Flecainide 2 μM considered the high levels of INa block observed at fast activation rates (~60 to 70% block at 10Hz, Moreno et al., 2011).*

Lines 482-491: *Different degrees of INa inhibition were considered, including high degrees of block observed at fast activation rates (Moreno et al., 2011). Accordingly, we observed an increase in flecainide efficacy with increasing INa block, particularly in atria with low G_{Na}. In patients with INa down-regulation, additional INa inhibition (i.e., flecainide) could pose a risk to the normal electrical conduction. However, the severe INa block exerted by flecainide in this study would only be expected at very high atrial rates (i.e., those observed during fibrillation). Under these high rates, flecainide application might even result in a complete loss of tissue excitability, as seen in atrial trabeculae after the application of compounds with a strong rate-dependent INa block (Wettwer et al., 2013). Conversely, a minimal INa block, and thus preserved conduction, would be expected during normal sinus rhythm (i.e., lower atrial rates).*

9. Fig 5: I'm having a hard time connecting the AF terminated vs not terminated ratios (e.g., 50 vs 38 for vernakalant) in panel B to the success rates in panel A. I'm assuming it has to do with the LVA levels but please provide info/clarification on this.

We appreciate the reviewer's comment. These ratios are indeed related to the LVA distributions, and we acknowledge they were not straightforward to connect without additional data. We have improved the figure and the text, to make it clearer. In short, we consider 100 ionic current profiles and 8 LVA distributions. We assume that an ionic current profile responds to a drug if AF is terminated in all LVA distributions in which the ionic current profile has sustained AF. This holds true if the ionic current profile has sustained AF in any number of LVA distributions. Thus, without knowing the exact number of LVA distributions in which the ionic current model sustained AF, it becomes challenging to connect this ionic current model in Figure 5B to the efficacy rate in Figure 5A. The text now reads:

Lines 381-388: *Figure 5B analyses the pharmacological outcome (i.e., AF termination vs. non-termination) of three drugs commonly used for rhythm control of AF according to the atrial ionic current profile (88 of the 100 profiles are included in the analysis since 12 failed to sustain AF, Figure 2). These ionic profiles have been grouped into three categories: profiles with a favorable drug response in all (100%) LVA distributions, in most of them (50-90% of LVA distributions), and in only a few (0-50%). Figure 5C infers about potential patient stratification by differentiating the ionic current characteristics of atria responding to one drug or another.*

Moreover, considerable improvements have been made in Figure 5B-C, for a clearer understanding of the comparisons.

Referee #2:

Major comments:

1. **Could the reviewers please provide more information on their implementation of imposing spiral wave reentry (Matene and Jacquemet 2012)? Beyond citing the original paper, it is important that readers understand how this was done in the current context. What were the initial conditions (e.g., how many phase singularities, direction of rotation, distance between phase singularities)? Adding this information is extremely important for reproducibility.**

We appreciate the reviewer's comment. An extended explanation has now been added to the manuscript:

Lines 173-181. *AF was induced in virtual atria by imposing spiral wave re-entries as the initial conditions of the simulation, and AF dynamics were analyzed for 7 s of activity (Matene and Jacquemet, 2012; Roney et al., 2022). Six spiral waves re-entries were imposed in the atria (Matene and Jacquemet, 2012), three in each atrial chamber. In the left atrium, one spiral wave was induced in the anterior wall, one in the posterior, and another in the inferior wall (equally distributed in space). Two spiral waves were induced in the venous portion of the right atrium, one around the proximity of the superior cava vein and one close to the inferior cava vein, and a third one in the inferior wall. The direction of rotation was clockwise for two spiral waves in each chamber and counter-clockwise in the third one, ensuring that adjacent re-entries rotated with opposing phase (Matene and Jacquemet, 2012). The sensitivity of AF induction by spiral-wave re-entries is addressed in the discussion.*

2. **Furthermore, related to the above point #1, please discuss the sensitivity of the rotor imposition methodology to aforementioned initial conditions? Is it possible that slight changes in initial phase singularity position might render a case inducible with AF when it was previously non-inducible (or vice versa?)**

We thank the reviewer for the interesting observation. We have observed that using a different set of phase singularities might change the duration of non-sustained episodes of AF. For instance, episodes lasting 2 s might last 3 s (or vice versa) if different phase singularities are used. However, episodes of sustained AF are mainly preserved regardless of the set of phase singularities, since the mechanisms maintaining AF were primarily modulated by the structural and ionic current substrate. An explanation is provided in the discussion:

Lines 180: *The sensitivity of AF induction by spiral-wave re-entries is addressed in the discussion.*

Lines 582-594: *The AF induction protocol imposed spiral wave re-entries as the initial condition for the simulation. Thus, slight changes in the number and location of these phase singularities might influence AF dynamics. However, we observed that the mechanisms maintaining AF were primarily modulated by the structural and ionic current substrate. As shown here and elsewhere (Matene and Jacquemet, 2012; Dasi et al., 2022), atria with high I_{CaL} presented high propensity for APD alternans and wavefront fractionation. Under this*

ionic current substrate, the authors (Matene and Jacquemet, 2012) observed that the complexity of AF dynamics was not controlled by the initial number of phase singularities. On the other extreme, rotors tended to anchor in structurally-remodeled substrate (i.e., LVA) for atria with low I_{CaL}. Similarly in this case, the initial position of spiral waves did not modulate the resulting AF dynamics. Most importantly, the bias associated with the exact selection of phase singularities was addressed by the comparative nature of this study. This is, the method chosen for inducing AF ensured that all scenarios tested shared comparable initiation conditions (Roney et al., 2022).

3. In Figure 2A, the authors show simulated ECG recordings and clinical ECG recordings. Although this figure is adapted from a prior publication, several clarifications and changes are needed.

We thank the reviewer for their comments. Answers are provided to each of the issues risen.

a. First, in this context, it is unclear exactly how the atrial-only clinical ECG recordings were derived. Are these invasive measurements or non-invasive measurements?

The atrial-only clinical ECG are derived from standard 12-lead ECGs (non-invasive measurements). This is explained in the original publication and a summary is included in the legend of Figure 2.

Lines 264-267: *The clinical ECG was recorded on the day of electrical cardioversion with a sampling frequency of 250 Hz. Postprocessing consisted of a zero-phase band-pass filter between 1 and 100 Hz, a 50-Hz notch filter and cancellation of ventricular signals: QRST-waves. For further details refer to Lankveld et al., 2016.*

b. What type of filtering was used to achieve this schematic?

The clinical ECG was taken from the original publication (figure attached here). We only took the precordial leads (V1-V6) after QRTS cancellation (highlighted with the red box). The contrast of the image was enhanced to accentuate the black lines of the ECG recording. No further modification was made.

Figure 1 (*Lankveld et al., 2016*). Computation of the electrocardiography (ECG)-derived parameters. The left panel shows 3 seconds of every lead before any filtering or preprocessing steps. The middle panel shows the same 3 seconds after 1- to 100-Hz band-pass filtering and QRST cancellation. The right panel shows the ECG- derived parameters computed on lead V1. The upper right panel shows the frequency-domain parameters computed on the frequency power spectrum. The bottom right panel shows the time-domain parameters.

- c. **Second, what are the respective time and voltage scales for both simulation and clinical signals? At minimum, these should be added.**

The time and voltage scales have now been added in Figure 2. Thanks for the comment.

- d. **Third, after examining the original version of the figure from Lankveld et al. side-by-side with the version used in this paper, it appears the authors accidentally flipped the time series data about the horizontal axis (i.e., the signals are left-right mirrored). This leads to the final point, which is that visual analysis suggests that there is not a solid relationship between the simulated and clinical ECGs. The way the data are presented might lead a naïve reader to assume this is some kind of quantitative validation, when the point (at best) seems to be that the signals are qualitatively similar-ish. The usefulness of the comparison is somewhat undermined by the fact that the authors accidentally flipped the signals from the original paper and the data still look basically "similar".**

Thanks for noticing the mistake of accidentally flipping the clinical ECG from the original publication. We have now corrected it. The conclusions we draw are still valid as we obtained simulated ECGs of AF that qualitatively resemble those obtained in human patients. Besides having the same voltage scale (which is of secondary importance, see below) the morphology and most importantly, the time scale (that relates to AF frequency) are the same. This evidences that our simulated AF shows the level of complexity that would be expected in ECGs from human patients. The voltage scale is of secondary importance since it is determined by countless variables (distance of the electrode respect to the heart, skin contact of the electrodes, ionic current profile of the patient, concrete time when the ECG is recorded, circadian rhythm, ECG post-processing, etc). Thus, further quantitative comparisons would be of limited value.

This has been made clear in the manuscript:

Lines 231-234. *Figure 2A illustrates a representative AF episode and the corresponding ECG. Simulated and clinical recordings shared similar morphology and complexity of fibrillatory-waves (f-waves), comparable time scales and AF frequencies, supporting the credibility of the simulation results.*

- e. **Further to point #3 above, this may also prompt a rethinking regarding the usefulness of presenting simulated ECG data in Figure 3.**

Given the explanation above, it is important to show the comparison between both ECGs as this lends credibility to the AF dynamics presented.

- 4. After all the ionic changes made (population model $\pm 50\%$); regional ionic differences; reduction in ion channel conductance for LVA regions; and drug associated changes for Figure 5) is it possible that some of these models were deranged into a non-physiologic state, and thus failed to reliably conduct altogether (i.e., following imposition of spiral waves no further excitation occurs)? In other words, could the authors please provide some validation that following ionic parameter changes the resulting functionality at the organ scale remains physiologically plausible?**

We thank the reviewer for the very interesting observation. The most severe case includes an ionic current profile presenting a scaling factor of -50% (resulting from the development of the population) and an additional block of 70% (after simulating drug effect). This could be the case of the sodium channel after flecainide simulation, in which the I_{Na} density, G_{Na} , would be reduced to 15% of its original value ($G_{Na} * 0.5 * 0.3 = G_{Na} * 0.15$).

This is still within the ranges tested in previous simulation studies: variations over a $\pm 100\%$ range (10.3389/fbioe.2017.00029) and between -100% to +200% of their original value (10.1016/j.hrthm.2016.08.028).

Moreover, the high G_{Na} block (60-70%) considered in this study tries to mimic the strong rate-dependent effects of Class Ic blockers (reference added in the main text):

Lines 210-211: *Flecainide 2 μ M considered the high levels of I_{Na} block observed at fast activation rates (~60 to 70% block at 10Hz, Moreno et al., 2011).*

We have observed that all ionic current models are able to propagate the stimulus at control (1Hz) pacing after virtual flecainide administration. Some of them (those responding to flecainide) are less excitable at fast pacing. In these cases of fast pacing, even experimental studies have observed complete loss of excitability after drug application.

This has been added in the text:

Lines 482-491: *Different degrees of I_{Na} inhibition were considered, including high degrees of block observed at fast activation rates (Moreno et al., 2011). Accordingly, we observed an increase in flecainide efficacy with increasing I_{Na} block, particularly in atria with low G_{Na} . In patients with I_{Na} down-regulation, additional I_{Na} inhibition (i.e., flecainide) could pose a risk to the normal electrical conduction. However, the severe I_{Na} block exerted by flecainide in this study would only be expected at very high atrial rates (i.e., those observed during fibrillation). Under these high rates, flecainide application might even result in a complete loss of tissue excitability, as seen in atrial trabeculae after the application of compounds with a strong rate-dependent I_{Na} block (Wettwer et al., 2013). Conversely, a minimal I_{Na} block, and thus preserved conduction, would be expected during normal sinus rhythm (i.e., lower atrial rates).*

- 5. Lines 495-515: Increased LVA infiltration in the left atria leading to rotors in the structurally-healthy right atria coexisting with rotors in areas of structural-remodeling**

is an interesting finding. Could it be possible that these right atrial rotors exist due to an imposed spiral wave in that region? In this case, a reentrant source arises from the induction method rather than any physiologic manner (i.e., triggered ectopy or downregulation of I_{CaL} and upregulation of I_{K1}, I_{to} and I_{NaK} as the authors describe).

We thank the reviewer for this observation. While the imposition of spiral waves re-entries helps with the stabilization of rotors, we have consistently observed in our studies that the electrophysiological differences in the right atrium (different APD, CV and anisotropy ratio in the crista terminalis, pectinate muscles and right atrial body) can by themselves initiate and perpetuate AF. This has also been observed in other studies (10.1113/jphysiol.2013.254987), that showed how the electrophysiological gradients between the crista terminalis and pectinate muscles increased tissue vulnerability to re-entry initiation and maintenance.

Nevertheless, since we have not induced AF by triggered activity but by imposing spiral waves, we have replaced “formation of rotors” by “stabilization of rotors”.

Line 540-541: *Several ionic current dysregulations contributed to the stabilization of the above-mentioned functional drivers, namely, the down-regulation of I_{CaL} and the up-regulation of I_{K1}, I_{to} and I_{NaK}.*

Minor comments:

1. **Lines 152-154: What adjustments were made to account for the regional ionic differences, and where did the information come from?**

The adjustments were described in our previous publication (10.3389/fphys.2022.966046). This has been added in the text:

Line 153-156: *The single-cell properties of each atrial cardiomyocyte model were assigned to the left atrial tissue and modified in right atrium, crista terminalis, pectinate muscles, left atrial appendage and atrio-ventricular rings (a detailed explanation can be found in Dasí et al., 2022).*

2. **Figure 2B: What was the rationale for grouping in an uneven manner? The manuscript would be strengthened if there were a clearly defined basis for determining the number of LVA distributions in each group.**

We thank the reviewer for their comment. To clarify this, a new grouping of the ionic current profiles has been implemented. This has been added in the main manuscript:

Lines 234-240. *Figure 2B characterizes the atrial ionic current substrate favoring AF maintenance. The 100 ionic current profiles included in this study have been grouped into three categories according to their proarrhythmic potential: profiles sustaining AF (>7s) for less than two LVA distributions (less than 25% of the LVA distributions studied), between two and six (25-75% of the LVA distributions) and for more than six LVA distributions (more than 75% of the LVA distributions). Figure 2C and 2D show the left atrium parcellation and the ionic current substrate needed for AF maintenance as the LVA extension increased.*

3. **Line 30: Clarify sentence to "pharmacological" or "drug-based" treatment.**

We thank the reviewer for their comment. This sentence has been clarified.

4. **Figure 3: Is there a reason why boxplots were used instead of violin plots?**

We thank the reviewer for their comment. The boxplots have been replaced by violin plots.

5. **Lines 256-267: The authors should consider qualifying the language in this paragraph as changes in ionic current dysregulation were not reliably identifiable for all distributions of LVA. For example, in low LVA burden cases GCaL and Gto did not appear to affect spectral entropy with statistical significance. Similar cases can be made for GNaK/sample entropy and GNa/relative harmonic energy.**

We thank the reviewer for their comment. This paragraph has been rewritten accordingly.

Line 285: *For some LVA distributions, high GCaL and low Gto increased the ECG spectral entropy.*

Line 289-290: Conversely, low GCaL and high Gto were sometimes associated with an elevation of the sample entropy

Line 292-293: High GNaK was also linked to high sample entropy for some LVA distributions.

Dear Mr Dasi,

Re: JP-RP-2023-284730R1 "What determines the optimal pharmacological treatment of atrial fibrillation? Insights from in-silico trials in 800 virtual atria" by Albert Dasi, Michael T.B. Pope, Rohan S. Wijesurendra, Tim R. Betts, Rafael Sachetto, Alfonso Bueno Orovio, and Blanca Rodriguez

Thank you for submitting your revised Research Article to The Journal of Physiology. It has been assessed by the original Reviewing Editor and Referees and has been well received. Some final revisions have been requested. Please advise your co-authors of this decision as soon as possible.

REVISION CHECKLIST:

- 'Potential Cover Art' for consideration as the issue's cover image

- Appropriate Supporting Information (Video, audio or data set: see https://jp.msubmit.net/cgi-bin/main.plex?form_type=display_requirements#supp).

We look forward to receiving your revised submission.

Yours sincerely,

Harold D Schultz
Senior Editor
The Journal of Physiology
<https://jp.msubmit.net>
<http://jp.physoc.org>
The Physiological Society
Hodgkin Huxley House
30 Farringdon Lane
London, EC1R 3AW
UK
<http://www.physoc.org>
<http://journals.physoc.org>

REQUIRED ITEMS FOR REVISION

-The contact information provided for the person responsible for 'Research Governance' at your institution is an author on this paper. Please provide an alternative contact who is not an author on this paper or confirm that the author whose email was provided has sole responsibility for research governance. This is the person who is responsible for regulations, principles and standards of good practice in research carried out at the institution, for instance the ethical treatment of animals, the keeping of proper experimental records or the reporting of results.

-You must start the Methods section with a paragraph headed Ethical Approval. If experiments were conducted on humans confirmation that informed consent was obtained, preferably in writing, that the studies conformed to the standards set by the latest revision of the Declaration of Helsinki, and that the procedures were approved by a properly constituted ethics committee, which should be named, must be included in the article file. If the research study was registered (clause 35 of the Declaration of Helsinki) the registration database should be indicated, otherwise the lack of registration should be noted as an exception (e.g. The study conformed to the standards set by the Declaration of Helsinki, except for registration in a database.). For further information see: <https://physoc.onlinelibrary.wiley.com/hub/human-experiments>

-The Journal of Physiology funds authors of provisionally accepted papers to use the premium BioRender site to create high resolution schematic figures. Follow this link and enter your details and the manuscript number to create and download figures. Upload these as the figure files for your revised submission. If you choose not to take up this offer we require figures to be of similar quality and resolution. If you are opting out of this service to authors, state this in the Comments section on the Detailed Information page of the submission form. The link provided should only be used for the purposes of this submission. Authors will be charged for figures created on this premium BioRender account if they are not related to this manuscript submission.

-Papers must comply with the Statistics Policy https://jp.msubmit.net/cgi-bin/main.plex?form_type=display_requirements#statistics

In summary:

-If n {less than or equal to} 30, all data points must be plotted in the figure in a way that reveals their range and distribution. A bar graph with data points overlaid, a box and whisker plot or a violin plot (preferably with data points included) are acceptable formats.

-If $n > 30$, then the entire raw dataset must be made available either as supporting information, or hosted on a not-for-profit repository e.g. FigShare, with access details provided in the manuscript.

- n clearly defined (e.g. x cells from y slices in z animals) in the Methods. Authors should be mindful of pseudoreplication.

- All relevant 'n' values must be clearly stated in the main text, figures and tables, and the Statistical Summary Document (required upon revision)
- The most appropriate summary statistic (e.g. mean or median and standard deviation) must be used. Standard Error of the Mean (SEM) alone is not permitted.
- Exact p values must be stated. Authors must not use 'greater than' or 'less than'. Exact p values must be stated to three significant figures even when 'no statistical significance' is claimed.
- Statistics Summary Document completed appropriately upon revision

-A Data Availability Statement is required for all papers reporting original data. This must be in the Additional Information section of the manuscript itself. It must have the paragraph heading "Data Availability Statement". All data supporting the results in the paper must be either: in the paper itself; uploaded as Supporting Information for Online Publication; or archived in an appropriate public repository. The statement needs to describe the availability or the absence of shared data. Authors must include in their Statement: a link to the repository they have used, or a statement that it is available as Supporting Information; reference the data in the appropriate section(s) of their manuscript; and cite the data they have shared in the References section. Whenever possible the scripts and other artefacts used to generate the analyses presented in the paper should also be publicly archived. If sharing data compromises ethical standards or legal requirements then authors are not expected to share it, but must note this in their Statement. For more information, see our Statistics Policy.

-Please include an Abstract Figure file, as well as the figure legend text within the main article file. The Abstract Figure is a piece of artwork designed to give readers an immediate understanding of the research and should summarise the main conclusions. If possible, the image should be easily 'readable' from left to right or top to bottom. It should show the physiological relevance of the manuscript so readers can assess the importance and content of its findings. Abstract Figures should not merely recapitulate other figures in the manuscript. Please try to keep the diagram as simple as possible and without superfluous information that may distract from the main conclusion(s). Abstract Figures must be provided by authors no later than the revised manuscript stage and should be uploaded as a separate file during online submission labelled as File Type 'Abstract Figure'. Please ensure that you include the figure legend in the main article file. All Abstract Figures should be created using BioRender. Authors should use The Journal's premium BioRender account to export high-resolution images. Details on how to use and access the premium account are included as part of this email.

EDITOR COMMENTS

Reviewing Editor:

The revision was largely satisfactory. A few issues and suggestions remain - particularly one regarding the comparison between simulation and ECG clinical data - that should be addressed.

Senior Editor:

The revised manuscript has been evaluated and found to be markedly improved. Both external reviewers have some remaining concerns that need to be addressed.

The Journal policy regarding reporting p values applies only when inferential statistics are used. The journal does also value the application of descriptive statistics to evaluate data when relevant. When done, please include effect sizes and confidence intervals.

REFEREE COMMENTS

Referee #1:

The authors have revised the manuscript and clarified several issues. I have one question about the revisions and a few minor suggestions.

Question:

Regarding the revisions made in response to point 5 in my original review: I'm not convinced about the importance of GCaL ("The ECG dominant frequency was mainly modulated by GK1 and GCaL"). Table 2 suggests a very limited role of GCaL as does comparing "Ionic current upregulation" to "Ionic current upregulation + ICaL downregulation" in the originally submitted figure 3. In the new figure 3 (A, bottom), the analysis points to a significant difference but this is for a very small subset of AF episodes (why?). If you compare all episodes with increased ICaL against those with decreased ICaL (regardless of all other conductance scalings), would there be a statistically significant effect?

Minor suggestions:

Regarding point 1 of my original review: Instead of "Sustained (>7s) AF episodes were subjected to", I think "Virtual atria with initial conditions that lead to sustained (>7s) AF episodes were subjected to" would make the process more clear to readers.

Regarding point 3 of my original review: I wasn't completely specific. I was referring to the clinical ECG and I appreciate the more detailed reference regarding this data. The addition to the text regarding your simulated ECG ("Only atrial electrical signals were simulated, so that ventricular cancellation (QRS- and T-wave removal) was not needed. ") is not necessary for my sake - I think it's clear that you are simulating the atria only.

Regarding Figs 2 and 5: the violin plots extend beyond the allowed range for variation (i.e., +-50%). I suggest chopping the violins at these bounds to make a clearer/fairer presentation.

Referee #2:

The authors have thoroughly revised the paper and the majority of my points have been adequately addressed. Only one point of contention remains, related to the comparison of simulated and clinical electrocardiographic data.

The statement "Simulated and clinical recordings shared similar morphology and complexity of fibrillatory-waves (f-waves), comparable time scales and AF frequencies, supporting the credibility of the simulation results" is simply not supported by the data presented. Imprecise language in this context could be very misleading. When the authors refer to "comparable time scales," that is only true in the micro scale as it pertains to the periodicity of the fibrillatory activity itself. More broadly speaking, it is in fact very unlikely that the simulated and clinical AF episodes have "comparable time scales" because the simulations only last 7 seconds, whereas in a patient an episode of AF lasting <30 seconds might not even be classified as clinically significant. Many paroxysmal AF episodes last hours or days, while patients with the persistent form of the disease are afflicted by incessant episodes that last >1 week by definition. Ergo, from this standpoint, it cannot possibly be said that the simulated and clinical AF episodes have "comparable time scales".

The authors have convinced me that it is worthwhile to include the ECG comparison at all, but in order to support the claims the authors are asserting to be true (i.e., there are broad similarities between the clinical and simulated signals), I believe (A) the comparison must be quantitative, and (B) the claims on similarity must be very carefully crafted to avoid any potential misinterpretation as described above. For (A), it would be adequate to annotate Fig. 2A with calculated values of dominant frequency (DF) *for these two sets of signals* - this should be straightforward, since the authors appear to have already done the calculation for the clinical ECG (DF=6.35; see response document). As an added benefit, this would improve the interpretability of the data presented in Fig. 3, since in the present manuscript the DF values are given for increased/decreased IK1 expression, but there is no control/baseline value given for the purpose of comparison.

END OF COMMENTS

EDITOR COMMENTS

Reviewing Editor:

The revision was largely satisfactory. A few issues and suggestions remain - particularly one regarding the comparison between simulation and ECG clinical data - that should be addressed.

Senior Editor:

The revised manuscript has been evaluated and found to be markedly improved. Both external reviewers have some remaining concerns that need to be addressed.

The Journal policy regarding reporting p values applies only when inferential statistics are used. The journal does also value the application of descriptive statistics to evaluate data when relevant. When done, please include effect sizes and confidence intervals.

REFEREE COMMENTS

Referee #1:

The authors have revised the manuscript and clarified several issues. I have one question about the revisions and a few minor suggestions.

Question:

- 1. Regarding the revisions made in response to point 5 in my original review: I'm not convinced about the importance of G_{CaL} ("The ECG dominant frequency was mainly modulated by G_{K1} and G_{CaL} "). Table 2 suggests a very limited role of G_{CaL} as does comparing "Ionic current upregulation" to "Ionic current upregulation + I_{CaL} downregulation" in the originally submitted figure 3. In the new figure 3 (A, bottom), the analysis points to a significant difference but this is for a very small subset of AF episodes (why?). If you compare all episodes with increased I_{CaL} against those with decreased I_{CaL} (regardless of all other conductance scalings), would there be a statistically significant effect?**

We appreciate the reviewer's comment.

In Figure 3 (A, bottom) there is a group represented by a small number of samples (i.e., $N=21$) because we are selecting a very specific subgroup of virtual patients that satisfy many conditions (i.e., elevated I_{Io} , elevated I_{Na} , elevated I_{NaK} , elevated I_{K1} and elevated I_{CaL}). Moreover, since AF is facilitated by reduced I_{CaL} , a limited number of virtual atria present both sustained AF and "elevated I_{CaL} ", even when no other condition needs to be satisfied.

After revision, we agree with the reviewer that I_{CaL} has a secondary role on AF dominant frequency. As per the reviewer comment, we have analyzed the influence of I_{CaL} alone on the dominant frequency (DF, Table). While statistical significance is achieved, small differences are observed in both subgroups:

	AF episodes G_{CaL} higher	AF episodes G_{CaL} lower	P-value
--	------------------------------	-----------------------------	---------

	than baseline (N=158)	than baseline (N=336)	
DF (median \pm IQR)	6.2 \pm 1.5	6.8 \pm 1.6	3.3e-11

The reason why the dominant frequency is increased with lower I_{CaL} density owes to the shortening of the re-entrant wavelength and thus, faster depolarization of re-entrant drivers. However, this only happens when reduced I_{CaL} is combined with other ionic current dysregulations, mainly increased I_{K1} . This has been clarified in the main text:

Line 302-313: *The ECG dominant frequency was mainly modulated by $IK1$, with other ionic currents having a cumulative effect (Table 2). Besides $IK1$ up-regulation, the general analysis identified a higher frequency in atria presenting increased Ito , $INaK$ and INa . Nevertheless, a thorough patient stratification demonstrated that atria with high Ito , $INaK$ and INa but control $IK1$, showed no significantly higher AF frequency than the overall population (Figure 3A). This is illustrated through two simulated ECG recordings during AF (Figure 3B), showing high and low frequency f-waves for $IK1$ up- and down-regulation, respectively. Similarly, while faster AF episodes were obtained in atria that additionally presented low $ICaL$, atria with increased $ICaL$ showed significantly higher AF frequency than the overall population if Ito , $IK1$, $INaK$, INa were up-regulated. As shown in Figure 2A, the fastest AF episodes were obtained for the combination of increased Ito , $IK1$, $INaK$, INa and decreased $ICaL$. This ionic current profile yielded the shortest tissue refractoriness (108 ± 10.3 ms), which enabled a fast activation of re-entrant AF drivers.*

Minor suggestions:

- Regarding point 1 of my original review: Instead of "Sustained (>7s) AF episodes were subjected to", I think "Virtual atria with initial conditions that lead to sustained (>7s) AF episodes were subjected to" would make the process clearer to readers.**

We appreciate the reviewer's comment. We believe that AF sustenance depends, not only about the initial conditions, but also about how the atrial characteristics interplay during the 7 seconds of activity. We have tried to rewrite the sentence according to the reviewer's suggestion:

Line 197: *Virtual atria with sustained (>7s) AF were subjected to...*

- Regarding point 3 of my original review: I wasn't completely specific. I was referring to the clinical ECG and I appreciate the more detailed reference regarding this data. The addition to the text regarding your simulated ECG ("Only atrial electrical signals were simulated, so that ventricular cancellation (QRS- and T-wave removal) was not needed.") is not necessary for my sake - I think it's clear that you are simulating the atria only.**

The sentence "*Only atrial electrical signals were simulated, so that ventricular cancellation (QRS- and T-wave removal) was not needed*" has been removed according to the reviewer's comments.

- Regarding Figs 2 and 5: the violin plots extend beyond the allowed range for variation (i.e., $\pm 50\%$). I suggest chopping the violins at these bounds to make a clearer/fairer presentation.**

We thank the reviewer's suggestion. We have tried to chop the violin plots, and below is provided a comparative example of the original (left) and the chopped version (right).

We believe that the different trends between the subgroups analyzed can be better appreciated in the original version (left). Moreover, since the samples are plotted inside the violin plot (black circles), the reader can perfectly see that no sample exceeds the $\pm 50\%$ bound.

Figure 2B. Variation of G_{to} , G_{K1} , G_{Ks} , G_{CaL} and G_{NaK} (with respect to baseline) yielding sustained AF for less than 25% of LVA distributions (light grey), 25-75% (dark grey) or more than 75% of LVA distributions (black). The original (left) and the chopped version at $\pm 50\%$ of variation (right).

Referee #2:

The authors have thoroughly revised the paper and the majority of my points have been adequately addressed. Only one point of contention remains, related to the comparison of simulated and clinical electrocardiographic data.

1. The statement "Simulated and clinical recordings shared similar morphology and complexity of fibrillatory-waves (f-waves), comparable time scales and AF frequencies, supporting the credibility of the simulation results" is simply not supported by the data presented. Imprecise language in this context could be very misleading. When the authors refer to "comparable time scales," that is only true in the micro scale as it pertains to the periodicity of the fibrillatory activity itself. More broadly speaking, it is in fact very unlikely that the simulated and clinical AF episodes have "comparable time scales" because the simulations only last 7 seconds, whereas in a patient an episode of AF lasting <30 seconds might not even be classified as clinically significant. Many paroxysmal AF episodes last hours or days, while patients with the persistent form of the disease are afflicted by incessant episodes that last >1 week by definition. Ergo, from this standpoint, it cannot possibly be said that the simulated and clinical AF episodes have "comparable time scales".

We appreciate the reviewer's comment. We agree with the reviewer, and the sentence has been re-written to avoid a misinterpretation of "comparable time scales".

Line 235-238: *Figure 2A illustrates a representative AF episode and the corresponding ECG. Simulated and clinical recordings shared similar morphology and complexity of fibrillatory-waves (f-waves), as well as comparable AF dominant frequency, supporting the credibility of the simulation results.*

2. The authors have convinced me that it is worthwhile to include the ECG comparison at all, but in order to support the claims the authors are asserting to be true (i.e., there are broad similarities between the clinical and simulated signals), I believe (A) the comparison must be quantitative, and (B) the claims on similarity must be very carefully crafted to avoid any potential misinterpretation as described above. For (A), it would be adequate to annotate Fig. 2A with calculated values of dominant frequency (DF) *for these two sets of signals* - this should be straightforward, since the authors appear to have already done the calculation for the clinical ECG (DF=6.35; see response document).

We have modified Figure 2A according to the reviewer's suggestion:

Figure 2A. Snapshot of the atrial transmembrane voltage map during AF and location of the atria within the torso (left). Simulated and clinical ECG during AF and dominant frequency (DF) of lead V1.

We believe that the aspect “(B) the claims on similarity must be very carefully crafted to avoid any potential misinterpretation” has been addressed in the Reviewer's previous comment.

3. As an added benefit, this would improve the interpretability of the data presented in Fig. 3, since in the present manuscript the DF values are given for increased/decreased I_{K1} expression, but there is no control/baseline value given for the purpose of comparison.

We appreciate the reviewer's comment. Figure 3 not only considers the DF values for increased/decrease I_{K1} expression but, in fact, there is a violin plot comprising the dominant frequency of all 494 AF episodes lasting longer than 7 s, where the entire spectrum of DF can be observed.

Dear Dr Dasí,

Re: JP-RP-2023-284730R2 "What determines the optimal pharmacological treatment of atrial fibrillation? Insights from in-silico trials in 800 virtual atria" by Albert Dasí, Michael T.B. Pope, Rohan S. Wijesurendra, Tim R. Betts, Rafael Sachetto, Alfonso Bueno Orovio, and Blanca Rodriguez

We are pleased to tell you that your paper has been accepted for publication in The Journal of Physiology.

NOTE-Please include the Abstract Figure legend text within the main article (doc.) file.

Authors should note that it is too late at this point to offer corrections prior to proofing. The accepted version will be published online, ahead of the copy edited and typeset version being made available. Major corrections at proof stage, such as changes to figures, will be referred to the Editors for approval before they can be incorporated. Only minor changes, such as to style and consistency, should be made at proof stage. Changes that need to be made after proof stage will usually require a formal correction notice.

Yours sincerely,

Harold D Schultz
Senior Editor
The Journal of Physiology
<https://jp.msubmit.net>
<http://jp.physoc.org>
The Physiological Society
Hodgkin Huxley House
30 Farringdon Lane
London, EC1R 3AW
UK
<http://www.physoc.org>
<http://journals.physoc.org>

P.S. - You can help your research get the attention it deserves! Check out Wiley's free Promotion Guide for best-practice recommendations for promoting your work at www.wileyauthors.com/eeo/guide. You can learn more about Wiley Editing Services which offers professional video, design, and writing services to create shareable video abstracts, infographics, conference posters, lay summaries, and research news stories for your research at www.wileyauthors.com/eeo/promotion.

IMPORTANT NOTICE ABOUT OPEN ACCESS: To assist authors whose funding agencies mandate public access to published research findings sooner than 12 months after publication, The Journal of Physiology allows authors to pay an Open Access (OA) fee to have their papers made freely available immediately on publication.

You can check if your funder or institution has a Wiley Open Access Account here: <https://authorservices.wiley.com/author-resources/Journal-Authors/licensing-and-open-access/open-access/author-compliance-tool.html>.

EDITOR COMMENTS

Reviewing Editor:

All the comments have been addressed satisfactorily. Congratulations!

Senior Editor:

Thank you for adequate revision of the manuscript. Congratulations on its acceptance.

REFEREE COMMENTS

Referee #1:

The authors have addressed my comments

Referee #2:

I have no further comments. The authors have satisfactorily addressed all points of concern from my standpoint.

2nd Confidential Review

03-Jul-2023